# A hexokinase isoenzyme switch in human liver cancer cells promotes lipogenesis and enhances innate immunity

Laure Perrin-Cocon [1,9], Pierre-Olivier Vidalain [1,9], Clémence Jacquemin[1], Anne Aublin-Gex [1],
Keedrian Olmstead[2], Baptiste Panthu[1,3], Gilles Jeans Philippe Rautureau [4], Patrice André[1], Piotr Nyczka [5],
Marc-Thorsten Hütt[5], Nivea Amoedo [6], Rodrigue Rossignol[6,7], Fabian Volker Filipp [2,8],
Vincent Lotteau [1,10✉] & Olivier Diaz [1,10✉]

During the cancerous transformation of normal hepatocytes into hepatocellular carcinoma (HCC), the enzyme catalyzing the first rate-limiting step of glycolysis, namely the glucokinase (GCK), is replaced by the higher affinity isoenzyme, hexokinase 2 (HK2). Here, we show that in HCC tumors the highest expression level of *HK2* is inversely correlated to *GCK* expression, and is associated to poor prognosis for patient survival. To further explore functional consequences of the GCK-to-HK2 isoenzyme switch occurring during carcinogenesis, *HK2* was knocked-out in the HCC cell line Huh7 and replaced by *GCK*, to generate the Huh7-*GCK*$^+$/*HK2*$^-$ cell line. HK2 knockdown and GCK expression rewired central carbon metabolism, stimulated mitochondrial respiration and restored essential metabolic functions of normal hepatocytes such as lipogenesis, VLDL secretion, glycogen storage. It also reactivated innate immune responses and sensitivity to natural killer cells, showing that consequences of the HK switch extend beyond metabolic reprogramming.

[1] CIRI, Centre International de Recherche en Infectiologie, Univ Lyon, Inserm, U1111, Université Claude Bernard Lyon 1, CNRS, UMR5308, ENS de Lyon, 21 Avenue Tony Garnier, Lyon F-69007, France. [2] Cancer Systems Biology, Institute for Diabetes and Cancer, Helmholtz Zentrum München, Ingolstädter Landstraße 1, München D-85764, Germany. [3] Univ Lyon, CarMeN Laboratory, Inserm, INRA, INSA Lyon, Université Claude Bernard Lyon 1, Hôpital Lyon Sud, Bâtiment CENS ELI-2D, 165 Chemin du grand Revoyet, Pierre-Bénite F-69310, France. [4] Université de Lyon, CNRS, Université Claude Bernard Lyon 1, ENS de Lyon, Centre de RMN à Très Hauts Champs (CRMN), FRE 2034, 5 rue de la Doua, Villeurbanne F-69100, France. [5] Department of Life Sciences and Chemistry, Jacobs University, Campus Ring 1, Bremen D-28759, Germany. [6] CELLOMET, Centre de Génomique Fonctionnelle de Bordeaux, 146 Rue Léo Saignat, Bordeaux F-33000, France. [7] Univ. Bordeaux, Inserm U1211, MRGM, Centre hospitalier universitaire Pellegrin, place Amélie Raba Léon, Bordeaux F-33076, France. [8] School of Life Sciences Weihenstephan, Technical University München, Maximus-von-Imhof-Forum 3, Freising D-85354, Germany. [9] These authors contributed equally: Laure Perrin-Cocon, Pierre-Olivier Vidalain. [10] These authors jointly supervised: Vincent Lotteau, Olivier Diaz. ✉email: vincent.lotteau@inserm.fr; olivier.diaz@inserm.fr

Hepatocellular carcinoma (HCC) is the most common liver cancer and the fourth leading cause of cancer-related death[1]. HCC is closely linked to chronic liver inflammation, chronic viral hepatitis, exposure to toxins, and metabolic dysfunction such as non-alcoholic steatohepatitis (NASH). HCC is of poor prognosis, and treatments are essentially based on surgical resection, liver transplantation or aggressive chemo and/or radiotherapy. In patients with advanced HCC, broad-spectrum kinase inhibitors are approved[2] but with limited benefit[3]. Effective personalized therapies are needed but their development is impeded by our poor understanding of molecular mechanisms underlying HCC onset and progression. Efforts to characterize the disease on the basis of etiology and outcomes revealed metabolic deregulation as a hallmark of HCC progression[4]. Indeed, metabolic remodeling is critically required for tumor growth, since bioenergetic requirements and anabolic demands drastically increase[5–7]. For instance, HCC cells have lost their ability to secrete very low-density lipoproteins (VLDL), a highly specialized function of hepatocyte and can only secrete low-density lipoproteins (LDL)-like lipoproteins, indicating a defective lipogenesis and/or lipoprotein assembly[8].

Metabolic reprogramming in cancer cells involves the modulation of several enzymes by oncogenic drivers[6]. Targeting these enzymes is now considered as a therapeutic strategy for several types of cancers[6]. Among these enzymes, hexokinase 2 (HK2) stands out because of its elevated or induced expression in numerous cancers, including HCC[9]. Hexokinases control the first rate-limiting step of glucose catabolism by phosphorylating glucose to glucose-6-phosphate (G6P), fueling glycolysis as well as glycogen, pentose phosphate and triglyceride synthesis. The human genome contains four genes encoding distinct hexokinase isoenzymes, named HK1 to HK4 (HK4 is also known as glucokinase or GCK), with distinct enzymatic kinetics and tissue distributions. A fifth putative hexokinase enzyme was recently discovered but has not been fully characterized yet[10]. A switch from GCK to HK2 isoenzymes is occurring during the transition from primary to tumor hepatocytes so that HCC cell lines express HK2 but no longer GCK. HK2 expression level has been correlated with disease progression and dedifferentiation of HCC cells[11]. When HK2 is artificially knocked-down in HCC cell lines, glycolysis is repressed, and tumorigenesis is inhibited while cell death increases[9]. In addition, hexokinase function extends beyond metabolism towards autophagy, cell migration, and immunity, suggesting that the GCK-to-HK2 isoenzyme switch has broader consequences than initially suspected[12–15]. Here, we analyzed transcriptomic data of HCC biopsies and correlated hexokinase isoenzyme expression level with patient survival. This led us to generate a new cellular model of human HCC expressing GCK instead of HK2. A comparative analysis of GCK+ vs HK2+ HCC cell lines provided a unique opportunity to look into HK isoenzyme-dependent metabolic features, lipoprotein production and resistance to immune signals of liver cancer cells.

## Results

**Relative expression level of GCK and HK2 in HCC patients.** Although an isoenzyme switch from GCK to HK2 has been observed during the carcinogenesis process[16], whether hexokinase isoenzymes expression is predictive of patient survival is unclear. We first analyzed the transcriptomes (RNA-seq data) of 365 HCC biopsies from The Cancer Genome Atlas (TCGA) database[17,18] (Supplementary Data 1). For each HK, the individual gene expression level was used to stratify patients into two subgroups according to Uhlen et al.[18] and overall survival in the two subgroups was determined using a Kaplan-Meier's estimator. Although HK1 or HK3 expression level were not associated to

patient survival rate (Fig. 1a), highest expression levels of HK2 as previously described[19] and lowest expression levels of GCK in the tumors were associated with a lower survival rate. We thus stratified patients based on the GCK/HK2 expression ratio to combine these two markers (Fig. 1b). When patients were stratified on the basis of HK2 or GCK expression levels, the median survival between the corresponding subgroups differed by 33.8 and 36.5 months, respectively (Fig. 1a). This difference reached 42.8 months when the stratification of patients was based on the GCK/HK2 ratio (Fig. 1b). This demonstrated that the GCK/HK2 ratio outperforms HK2 or GCK expression alone as predictor of patient survival. Finally, correlation coefficients between patient survival in months and HK2 or GCK expression level were determined. For this, we only considered the subset of 130 patients for whom the period between diagnosis and death is precisely known (uncensored data), and performed a Spearman's rank correlation test (Fig. 1c). Patient survival was positively correlated to GCK expression but inversely correlated to HK2 expression in line with the Kaplan-Meier analysis. In addition, GCK and HK2 expression tends to be inversely correlated in tumor samples (Fig. 1c). Therefore, there is a trend for mutual exclusion of GCK and HK2 expression in HCC tumors, and this profile is associated to clinical outcome.

**Engineering a cellular model of the hexokinase isoenzyme switch.** To decipher functional consequences of GCK or HK2 expression in a HCC model, we restored GCK expression by lentiviral transduction in the reference HCC cell line Huh7, and knocked-out the endogenous HK2 gene by CRISPR/Cas9. The exclusive expression of HK2 and GCK in Huh7 and Huh7-GCK+/HK2− cell lines, respectively, was validated, while HK1 and HK3 were not expressed (Fig. 2a and Supplementary Fig. 1). The hexokinase activity in the presence of increasing concentration of glucose was determined in protein lysates from the two respective cell lines. Hexokinase activity in Huh7 lysate reached its maximum at low glucose concentration, presenting a saturation curve according to Michaelis–Menten kinetics (Fig. 2b). In contrast, the hexokinase activity in Huh7-GCK+/HK2− lysates followed a pseudo-allosteric response to glucose[20,21]. Thus, the expected HK2 and GCK activities were observed in the Huh7 and Huh7-GCK+/HK2− cells respectively. The cell proliferation capacity remained identical between the two cell lines (Supplementary Fig. 2). We then compared the genome edited Huh7-GCK+/HK2− and the parental Huh7 cell lines at a transcriptomic, metabolic and immunological level.

**Transcriptomic data revealed extended modifications of metabolic connections in Huh7-GCK+/HK2−.** Transcriptomic profiles of Huh7 and Huh7-GCK+/HK2− cells were determined by next generation sequencing (Supplementary Data 2). Overall, 4.2% of the gene transcripts were reduced and 6.9% were induced in Huh7-GCK+/HK2− compared to Huh7 (Fig. 2c; | fold-change (FC) | > 2 and p value<0.05). We first determined the metabolic consequences of the HK isoenzyme switch by mapping the differentially expressed genes onto the well-established bipartite metabolic network Recon2, connecting gene products and metabolites (Supplementary Figs. 3–5)[22,23]. After trimming highest-degree metabolites as currency metabolites, clusters of genes that are both differentially expressed and connected by common metabolites emerged. Interestingly, we found that across a wide range of analysis parameters, including varying rates of currency metabolites and gene expression fold-change, the differentially expressed metabolic genes are substantially better connected than expected by chance (Fig. 2d). This highlights the specificity of the transcriptomic changes with respect to metabolic pathways. The spanned network presented in Fig. 2e corresponds

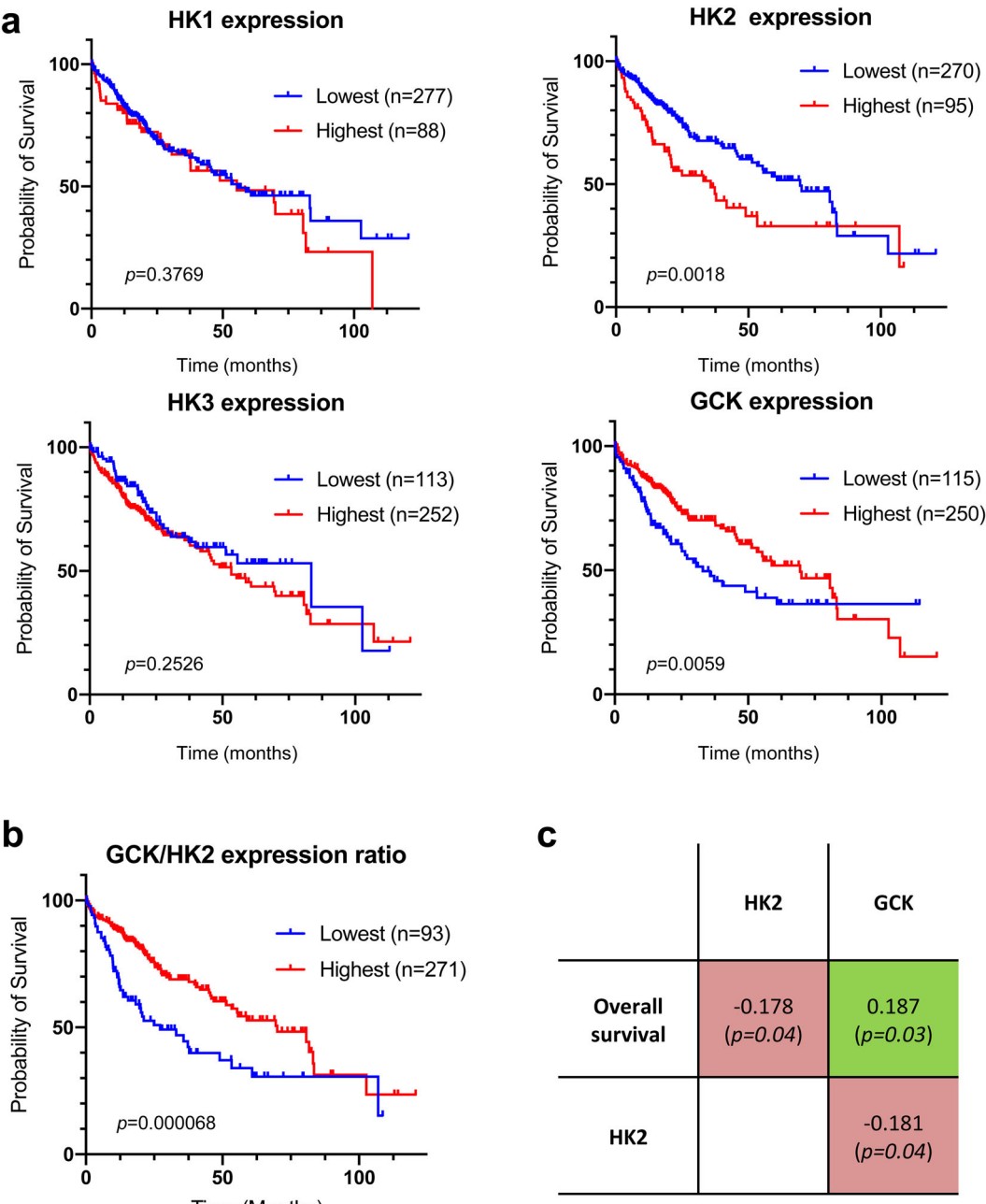

**Fig. 1 Correlation between hexokinase expression levels in HCC tumors and patient survival. a** Kaplan–Meier estimates of the survival of HCC patients depending on the expression of *HK1*, *HK2*, *HK3* and *GCK* (*HK4*) genes in tumor biopsies (*n* = 365; diploid samples; TCGA expression data retrieved from cBioPortal; Firehose Legacy)[74,75]. Duplicate analyses from the same patient were removed as well as patients who died when biopsied (overall survival=0 months or not specified). Optimal stratification based on highest and lowest gene expression values was determined using Protein Atlas database[18]. **b** Same as above but patients were stratified based on the *GCK/HK2* gene expression ratio. The stratification showing the lowest p value when comparing subgroups of patients with the highest to the lowest *GCK/HK2* expression ratio is displayed. Patient TCGA-DD-AAE9 exhibiting undetectable levels of *GCK* and *HK2* was removed from this analysis as the *GCK/HK2* ratio could not be calculated. **c** Correlations between patient survival, *GCK* expression and *HK2* expression. Spearman's rank correlation test on the subset 130 patients for whom the period between diagnosis and death is precisely known (uncensored data).

to a stringent fold-change threshold for transcriptomic data ($\log_2(|FC|) > 3$) while removing 2 percent of highest-degree currency metabolites. This network shows connected components within glycolysis, but also across distant modules including the gamma-aminobutyric acid (GABA) shunt (ALDH5A1), urea cycle (CPS1, OTC), glycogen metabolism (GYS1, GYS2, AGL) and lipid synthesis (GPAM, AGPAT4, DGKG, CDS1, A4GALT) or degradation (ACADL, HSD17B4, AMACR). This analysis

highlights the global impact of the HK isoenzyme switch that spreads beyond glycolysis across distant connected metabolic modules.

Enrichment of molecular and cellular functions in differentially expressed genes was also analyzed using Ingenuity Pathway Analysis (IPA). This revealed that cellular movement and lipid metabolism were the most affected functions (Table 1 and Supplementary Data 3). A closer look at these annotations

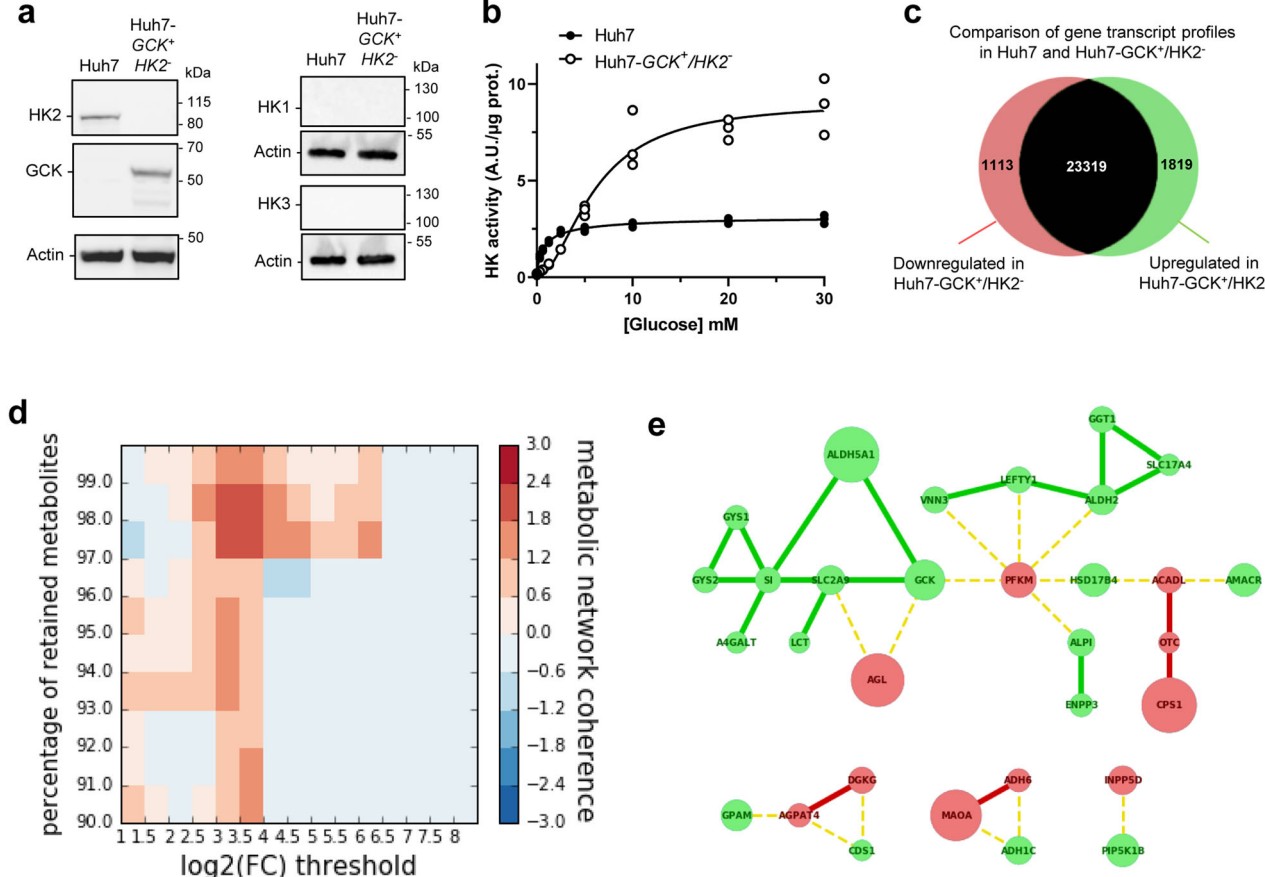

**Fig. 2 Hexokinase isoenzyme switch in Huh7 cells induces extended modifications of metabolic connections. a** Western-blot analysis of HK1, HK2, HK3 and GCK expression in Huh7 and Huh7-*GCK*+/*HK2*−. **b** Hexokinase activity in homogenates of Huh7 and Huh7-*GCK*+/*HK2*− cells. Means ± SEM are presented ($n = 3$). **c** Number of genes changing their expression pattern in Huh7 and Huh7-*GCK*+/*HK2*− cells (see Supplementary Data 2 for details). **d** Heatmap showing clustering enrichment scores of the networks obtained when mapping differentially expressed genes to the human metabolic model Recon2. Clustering enrichment scores from the highest in red to the lowest in blue were calculated for different gene expression thresholds ($Log_2|FC|$) and percentages of retained currency metabolites. **e** Gene network corresponding to the maximal clustering enrichment score ($Log_2|FC| > 3$; removed currency metabolites = 2%). The transcription of nodes in green was upregulated and those in red downregulated in Huh7-*GCK*+/*HK2*− compared to Huh7 cells. Plain edges mark co-regulation between nodes and broken edges inverse regulation at the transcriptional level.

**Table 1 Analysis of differentially expressed genes in Huh7 and Huh7-*GCK*+/*HK2*− using gene set enrichment analysis (|FC| > 2 with a *p* value <0.05).**

| Molecular and cellular function[a] | *p* value range | Number of genes involved |
|---|---|---|
| Cellular movement | $7.68 \times 10^{-6}$–$4.66 \times 10^{-25}$ | 701 |
| Lipid metabolism | $2.00 \times 10^{-6}$–$3.12 \times 10^{-14}$ | 414 |
| Molecular transport | $5.63 \times 10^{-6}$–$3.12 \times 10^{-14}$ | 361 |
| Small molecule biochemistry | $5.63 \times 10^{-6}$–$3.12 \times 10^{-14}$ | 462 |
| Protein synthesis | $1.87 \times 10^{-6}$–$3.42 \times 10^{-14}$ | 199 |

[a]Top-five enriched molecular and cellular functions are presented.

pointed to differences in the migratory capacities (Table 2) as well as lipid concentration and synthesis (Table 3). The migratory capacities of Huh7 and Huh7-*GCK*+/*HK2*− were compared using transwell-migration cell assays (Fig. 3a, b). Results showed a higher migratory capacity of Huh7-*GCK*+/*HK2*− cells, in line with Kishore M. et al showing that GCK expression induced by pro-migratory signals controls the trafficking of CD4+/CD25+ /FOXP3+ regulatory T (T$_{reg}$) cells[15]. To validate the differences in lipid metabolism highlighted by the transcriptomic analysis, intracellular content in neutral lipids was first determined with lipophilic dyes. As assessed by Oil-Red-O or BODIPY staining, an accumulation of intracellular neutral lipids was observed in Huh7-*GCK*+/*HK2*− in comparison to Huh7 (Fig. 3c, d). The accumulation of neutral lipids in Huh7 expressing both *HK2* and *GCK* indicates that the phenotype is driven by *GCK* expression and not by *HK2* knockdown (Supplementary Fig. 6). Lipid accumulation upon *GCK* expression was also observed in Huh6 hepatoblastoma cells but not in epithelial kidney Vero cells, indicating that this phenomenon occurs in metabolically relevant cells (Supplementary Fig. 6).

**Differential lipid metabolism in Huh7 and Huh7-$GCK^+/HK2^-$.** The intracellular lipid content of the two cell lines was further analyzed. In Huh7-$GCK^+/HK2^-$, an enrichment in phosphatidylcholine, cholesterol, triglycerides (TG) and free fatty acids was observed compared to Huh7 (Fig. 4a). One major function of hepatocytes is to secrete triglyceride-rich VLDL and this function is altered in HCC cells that secrete smaller lipoproteins with the density of LDL[24,25]. The secretion of lipids and lipoproteins by both cell lines was analyzed after a 24h-culture in the absence of fetal calf serum (FCS) to exclude any participation of exogenous lipids in the production of lipoproteins. Huh7-$GCK^+/HK2^-$ secreted more free fatty acids than Huh7 while secretion of cholesterol and TG remained unchanged (Fig. 4b). However,

under the same conditions, the secretion of apolipoprotein B (ApoB) by Huh7-$GCK^+/HK2^-$ was reduced compared to Huh7. Since ApoB is a non-exchangeable protein with only one copy in VLDL and LDL particles, an elevated TG/ApoB ratio indicates that ApoB$^+$-lipoproteins secreted by Huh7-$GCK^+/HK2^-$ cells are enriched in TG compared to those secreted by Huh7 (Fig. 4c). This was confirmed by the ApoB distribution in density gradient fractions. As expected, lipoproteins secreted by Huh7 sediment at the density of LDL, while those secreted by Huh7-$GCK^+/HK2^-$ (Fig. 4d) match the density of VLDL found in human plasma or secreted by primary human hepatocytes in culture[26,27]. This indicates that GCK expression is essential for the VLDL assembly/secretion pathway and could explain the loss of this crucial metabolic pathway in hepatoma cells expressing HK2 instead of GCK[28].

**Differential activity of the tricarboxylic acid cycle (TCA) in Huh7 and Huh7-$GCK^+/HK2^-$.** We observed that GCK expression increased the intracellular content in lipids, resulting in accumulation of lipid droplets and secretion of VLDL. A rewiring of cellular metabolism towards energy storage in Huh7-$GCK^+/HK2^-$ was thus suspected and confirmed by the accumulation of glycogen, creatine and creatine-P (Fig. 5a, b), a feature of functional hepatocytes. To further determine the consequences of replacing HK2 by GCK, we quantified prominent intracellular metabolites via gas chromatography coupled to triple-quadrupole (QQQ) mass spectrometry (GC-MS). Figure 5c shows relative intracellular quantities of metabolites that are significantly different between Huh7 and Huh7-$GCK^+/HK2^-$. Among differentially represented metabolites, higher levels of glucose, glycerol-3-phosphate and lactic acid were detected in Huh7-$GCK^+/HK2^-$ cells. Several intermediates of the TCA cycle (succinic acid, fumaric acid, alpha-ketoglutaric acid), and metabolites directly connected to it (GABA, glutamic acid, glutamine, aspartic acid) were also differentially present between the two cell lines. This supports a modulation of central carbon metabolism at both the level of glycolysis and TCA cycle. This led to investigate glucose catabolism in further details. Glucose consumption and stable

**Table 2 Top-five ranked IPA-annotations associated to 'cellular movement'.**

| Cellular movement functional annotations | p value | Number of genes involved |
|---|---|---|
| Migration of cells | $4.66 \times 10^{-25}$ | 585 |
| Cell movement | $3.48 \times 10^{-24}$ | 642 |
| Cell movement of blood cells | $1.52 \times 10^{-18}$ | 276 |
| Leukocyte migration | $1.59 \times 10^{-18}$ | 274 |
| Invasion of cells | $5.33 \times 10^{-17}$ | 306 |

**Table 3 Top-five ranked IPA-annotations associated to 'lipid metabolism'.**

| Lipid metabolism functional annotations | p value | Number of genes involved |
|---|---|---|
| Concentration of lipid | $3.12 \times 10^{-14}$ | 256 |
| Synthesis of lipid | $2.37 \times 10^{-10}$ | 233 |
| Fatty acid metabolism | $4.31 \times 10^{-10}$ | 166 |
| Quantity of steroid | $4.48 \times 10^{-09}$ | 138 |
| Concentration of cholesterol | $6.58 \times 10^{-08}$ | 94 |

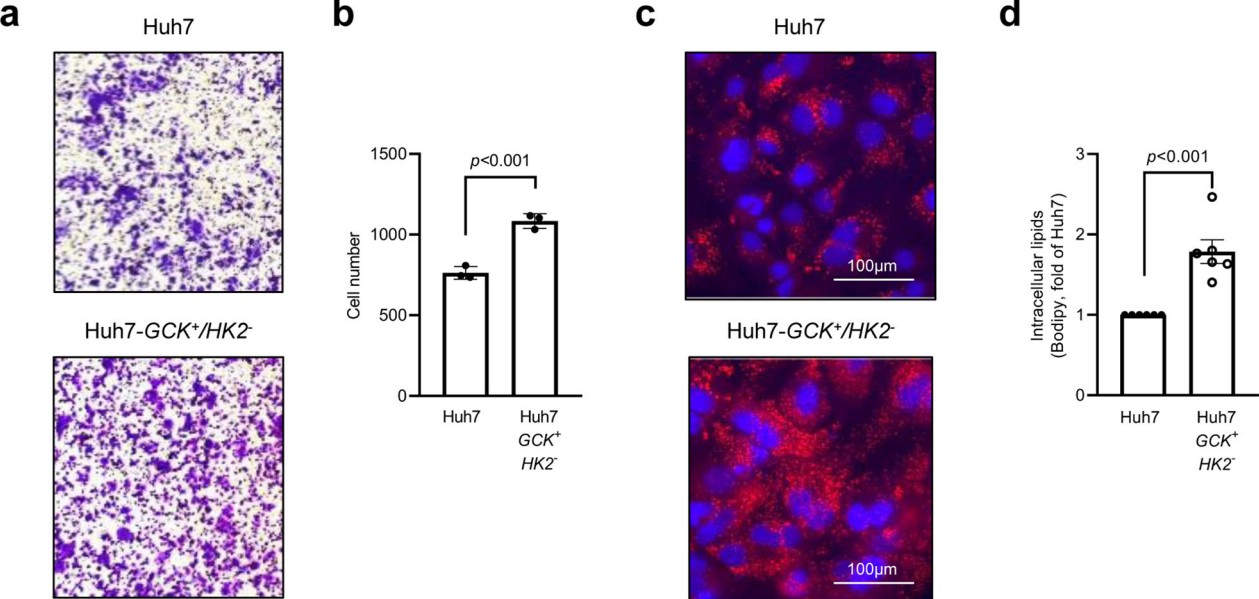

**Fig. 3 Huh7-$GCK^+/HK2^-$ cells have a higher migration capacity and lipid droplets content. a, b** Results of transwell-migration tests. **a** Representative images and **b** count of migrating cells ($n = 3$). **c** Oil Red-O staining of lipid droplets (red) with nucleus counterstaining (blue). **d** Quantification of intracellular lipids by FACS after BODIPY staining ($n = 6$). Means ± SEM are indicated and $p$ values were determined by Student's $t$-test.

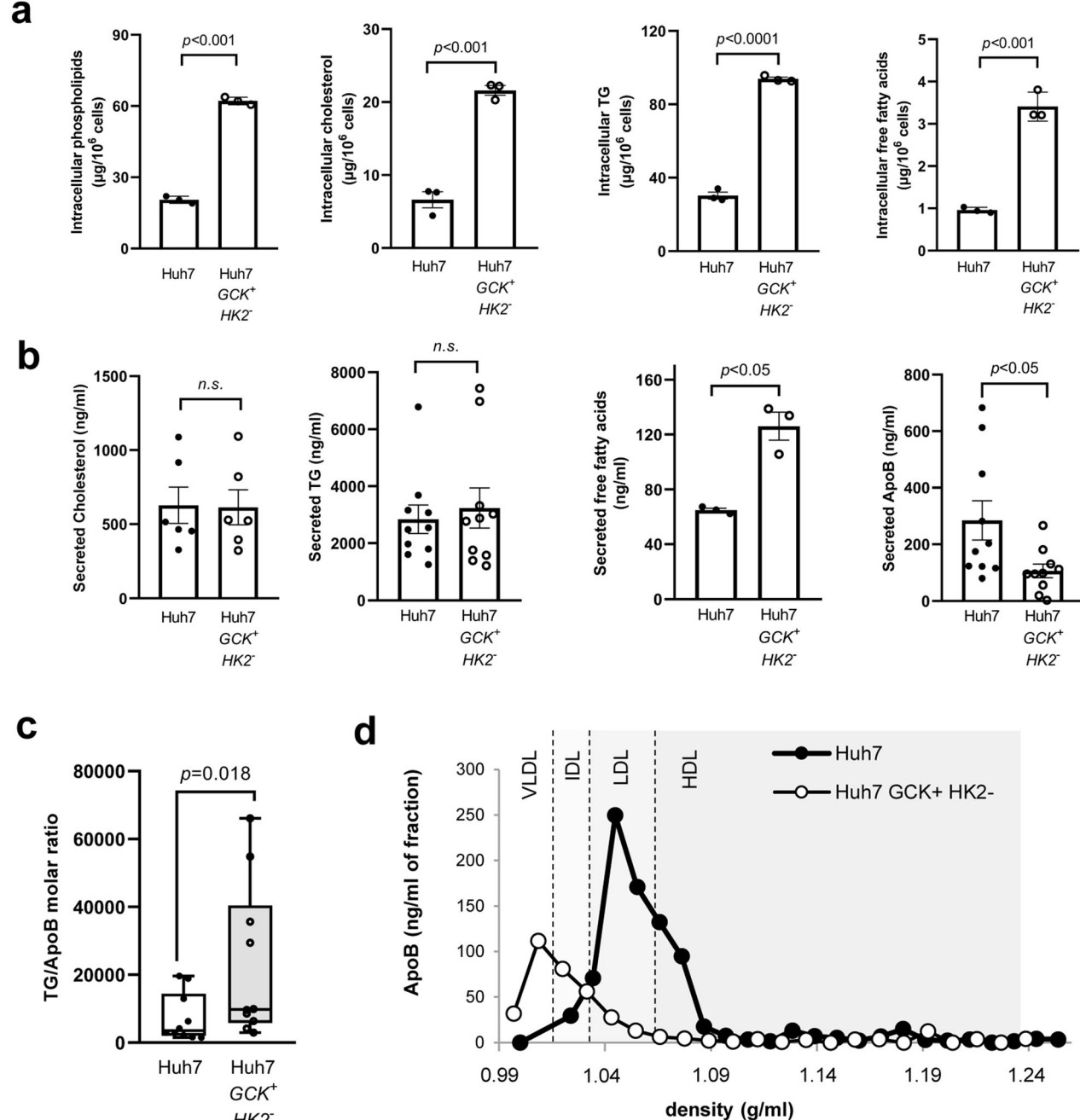

**Fig. 4 Lipogenesis and very-low-density lipoproteins (VLDL) secretion are restored in Huh7-*GCK⁺/HK2⁻* cells. a** Quantification of intracellular lipids in total cell extracts of Huh7 and Huh7-*GCK⁺/HK2⁻* cells ($n = 3$). **b** Lipids and ApoB secretions in supernatants of cells cultured 24 h without FCS ($n = 6$ for Cholesterol, $n = 3$ for FFA and $n = 10$ for TG and ApoB). **c** TG/ApoB molar ratio calculated from quantifications determined in **b** ($n = 10$). **d** Supernatants of Huh7 and Huh7-*GCK⁺/HK2⁻* were analyzed by ultracentrifugation on iodixanol density gradients. ApoB was quantified in each fraction by ELISA (one representative experiment). Presented data correspond to means ± SEM of indicated number of independent experiments and $p$ values were determined by Student's t-test.

isotope incorporation from [U-¹³C]-glucose into pyruvate were both increased in Huh7-*GCK⁺/HK2⁻* compared to Huh7 cells (Fig. 5d, e). This increased glycolytic flux together with a reduced lactate secretion (Fig. 5d) is likely to account for the elevation of lactate levels and suggest that the increased pyruvate production essentially fuels mitochondrial TCA cycle in Huh7-*GCK⁺/ HK2⁻* cells.

Pyruvate entering the mitochondria downstream of glycolysis can be either oxidized by pyruvate dehydrogenase (PDH), producing acetyl-CoA, or converted into oxaloacetate (OAA) by pyruvate carboxylase (PC). Acetyl-CoA and OAA are then combined in the TCA cycle to form citrate. *De novo* lipogenesis requires citrate egress from the TCA cycle to serve as a precursor of cytosolic acetyl-CoA for further synthesis of fatty acids. In Huh7-*GCK⁺/HK2⁻* cells, we observed both an increased activity of PC (Fig. 5f) without changes in protein expression (Fig. 5g and Supplementary Fig. 7a) and an increased phosphorylation of pyruvate dehydrogenase (PDH), which is indicative of a reduced activity of this enzyme (Fig. 5h and Supplementary Fig. 7b). This is consistent with the increased expression of the PDH kinase

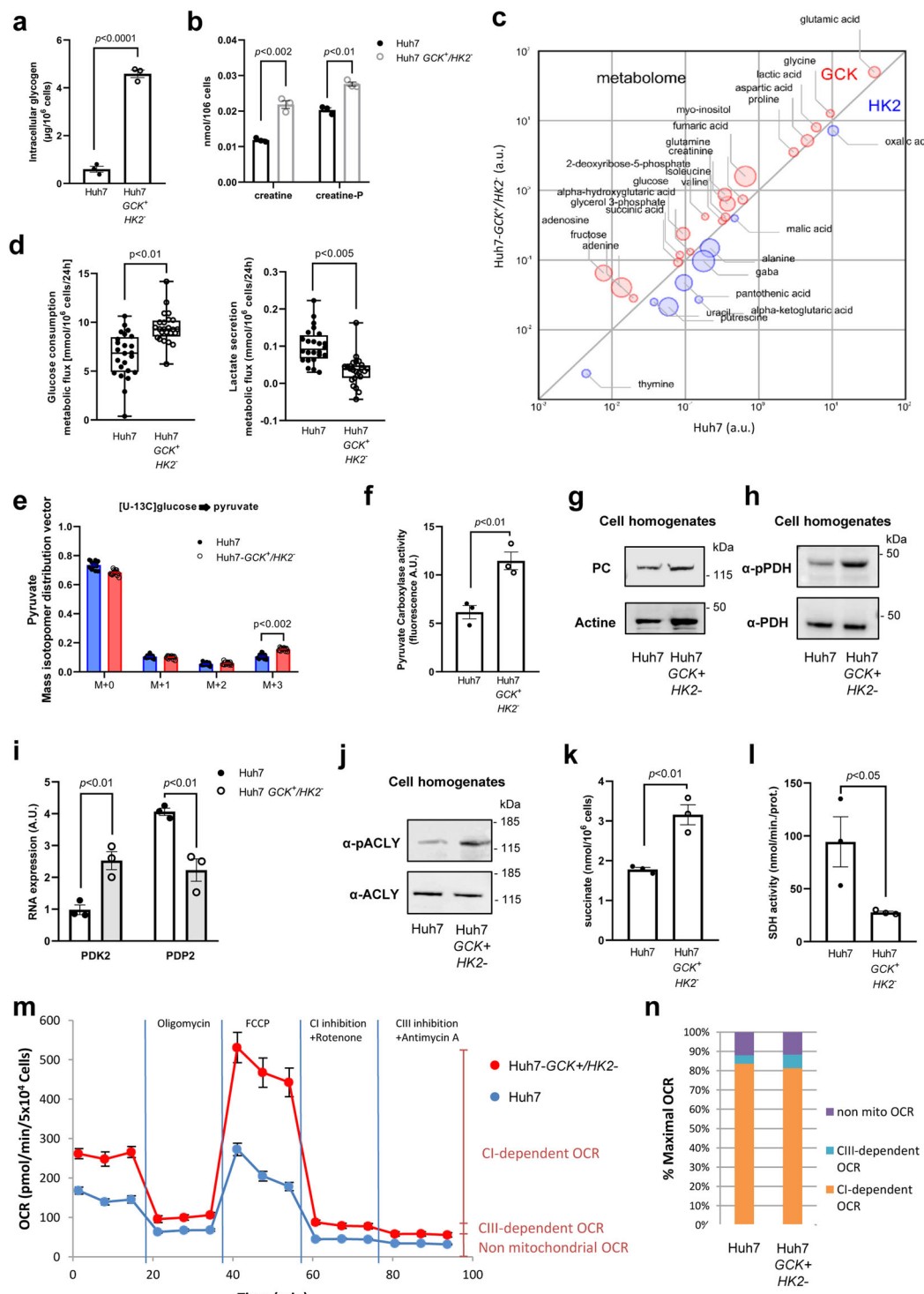

PDK2 and the decreased expression of the PDH phosphatase PDP2 in Huh7-$GCK^+$/$HK2^-$ cells that regulate the PDH phosphorylation state (Fig. 5i). A rebalanced usage of pyruvate in Huh7-$GCK^+$/$HK2^-$ cells maintains a functional TCA cycle and supports lipogenesis. In Huh7-$GCK^+$/$HK2^-$ cells, we also observed an increased phosphorylation of ATP citrate lyase (ACLY), the first enzyme of the fatty acid synthesis pathway, indicating an enhanced activity of this enzyme (Fig. 5j and Supplementary Fig. 7c). This reaction also regenerates OAA in the cytosolic compartment. Interestingly, transcriptomic data show that PCK1 which converts OAA to phosphoenolpyruvate

(PEP), is overexpressed in Huh7-$GCK^+$/$HK2^-$ cells compared to Huh7 (FC = 32).

A shift from pyruvate oxidation to carboxylation is observed in cancer cells where succinate dehydrogenase (SDH) is inactivated by mutation and OAA can only be generated through PC activity[29]. SDH inhibition leads to succinate accumulation, especially in activated immune cells[30]. Interestingly, higher levels of succinate and a reduced activity of SDH were measured in Huh7-$GCK^+$/$HK2^-$ compared to Huh7 cells (Fig. 5k, l). Even though SDH is also part of the complex II of the mitochondrial respiratory chain, we observed that the overall oxygen

**Fig. 5 TCA rewiring after hexokinase isoenzyme switch in Huh7 cells. a** Glycogen quantification. **b** Creatinine and creatinine-P quantification. **c** This bubble chart compares intracellular metabolomes of Huh7 and Huh7-$GCK^+/HK2^-$ cells. Metabolite pool sizes larger in Huh7 are indicated in blue, whereas the one larger in Huh7-$GCK^+/HK2^-$ are shown in red. The size of bubbles inversely scales with p values between $5.10^{-2}$ and $1.10^{-17}$ of differential metabolomics responses. **d** Metabolic fluxes for overall glucose consumption and lactate secretion by Huh7 and Huh7-$GCK^+/HK2^-$ cells. Indicated values correspond to differences in glucose or lactate concentrations in extracellular culture medium before and after 24 h of culture. **e** Mass isotopomer distribution vector of pyruvate in cells cultured with [U-$^{13}$C]-glucose. Presented data correspond to $n = 24$ (**c, d**) or $n = 16$ (**e**) acquired spectra from $N = 6$ and $N = 4$ independent specimens, respectively. **f** Pyruvate carboxylase (PC) activity determined in cell homogenates. **g** Western-blot analysis of PC expression in Huh7 and Huh7-$GCK^+/HK2^-$ cells. **h** Western-blot analysis of pyruvate dehydrogenase (PDH) E1-alpha subunit phosphorylation at Ser293. **i** RNA-seq quantification of pyruvate dehydrogenase kinase 2 (PDK2) and pyruvate dehydrogenase phosphatase 2 (PDP2) (BH adjusted p value<0.05 from transcriptomic data). **j** Western-blot analysis of ATP-citrate Lyase (ACLY) phosphorylation at Ser455. **k** Succinate quantification in cell homogenates. **l** Succinate dehydrogenase (SDH) activity determined in cell homogenates. **m** Oxygen consumption rate (OCR) in Huh7 and Huh7-$GCK^+/HK2^-$ cells was determined with a Seahorse analyzer before and after the addition of oligomycin (Complex V inhibitor), FCCP (uncoupling agent), rotenone (Complex I inhibitor) and antimycin A (Complex III inhibitor) ($n = 5$). **n** Non-mitochondrial, complex I-dependent and complex III-dependent maximal OCR were calculated from m. Except otherwise indicated, data correspond to means ± SEM of 3 independent experiments and p values were determined by Student's t-test.

consumption was increased in Huh7-$GCK^+/HK2^-$ (Fig. 5m) with increased basal and maximal respiration, ATP production and spare respiration capacity (Supplementary Fig. 8). Functional analysis of the respiratory chain showed that oxygen consumption in Huh7 and Huh7-$GCK^+/HK2^-$ cells was mainly dependent on complex I activity (Fig. 5m, n). Thereby, the HK isoenzyme switch rewired the TCA cycle promoting carboxylation of pyruvate into OAA in the presence of a reduced SDH activity and increased respiration through complex I.

**Restored innate immune sensitivity in Huh7-$GCK^+/HK2^-$.** Lipid accumulation in hepatocytes is incriminated in hepatic inflammation[31] and TCA cycle rewiring is associated with innate immunity activation[32]. These two events were observed when replacing HK2 by GCK in Huh7 cells, questioning the immune status of these cells and their sensitivity to antitumor immunity. The functional analysis of gene ontology (GO) terms associated to differentially expressed transcripts revealed an enrichment in terms related to the regulation of innate immunity. The gene signature associated with type-I interferon (IFN) signaling pathway scored among the top enriched terms of upregulated transcripts in Huh7-$GCK^+/HK2^-$ cells (Supplementary Fig. 9). Within the 91 gene members of this GO term, 20 transcripts of Type I-IFN signaling were significantly up-regulated in Huh7-$GCK^+/HK2^-$ compared to Huh7 (Fig. 6a). This includes interferon regulatory factors (*IRF1*, *IRF3* and *IRF9*), IFN-stimulated genes (ISGs) such as *ISG15*, *MX1*, *OAS1*, *OAS3*, *RNaseL* and signaling intermediates such as *IKBKE* coding for IKKε (Fig. 6b). The chaperon *HSP90AB1*, which is involved in the phosphorylation and activation of STAT1, was also induced[33]. In contrast, two genes were significantly down-regulated in Huh7-$GCK^+/HK2^-$ compared to Huh7, the RNaseL inhibitor *ABCE1*, and *TRIM6*, an E3 ubiquitin-protein ligase regulating IKKε. Among the members of the RIG-I-like receptor (RLR) family involved in immune sensing and antitumor defense[34], the expression of *IFIH1* (also known as *MDA5*) was slightly increased in Huh7-$GCK^+/HK2^-$, while *DDX58* (also known as *RIG-I*) itself remained unchanged. Such identified gene sets suggest that Huh7-$GCK^+/HK2^-$ cells may be better equipped than Huh7 cells to respond to innate immune signals. Thus, we compared the innate immune response of the two cell lines when stimulated by RIG-I or MDA5 ligands, known to sense different double-stranded RNA (dsRNA) in viral immune recognition. In order to quantify cellular activation by dsRNA, immuno-stimulatory challenge assays were conducted with triphosphate-hairpin RNA (3p-hpRNA) or polyinosinic-polycytidylic acid (poly(I:C)). Interferon-sensitive response element (ISRE)-dependent transcription was efficiently induced by these RLR ligands in Huh7-

$GCK^+/HK2^-$ cells whereas Huh7 response was very limited even at high doses of immuno-stimulatory ligands (Fig. 6c).

To investigate whether this differential sensitivity to RLR ligands is linked to GCK expression or HK2 knockout, we used Huh7 cells transduced for GCK expression so that they express both HK2 and GCK (Huh7-$GCK^+/HK2^+$). In these cells, the response to RIG-I ligation did not differ from that of Huh7 cells suggesting that GCK expression alone is not sufficient to restore immune sensitivity (Fig. 6d). When HK2 expression was repressed in these cells with a shRNA (Huh7-$GCK^+/HK2^-$Sh) (see Supplementary Fig. 10 showing 95% extinction of HK2 protein), ISRE response to RIG-I signaling was restored to a level similar to that observed in Huh7-$GCK^+/HK2^-$ cells (Fig. 6d). This is pointing towards HK2 as a negative regulator of RLR signaling in HCC cells and suggests that the GCK-to-HK2 isoenzyme switch during malignant transformation of hepatocytes is accompanied by a reduced sensitivity to innate immune signals. The higher sensitivity to RLR ligands of Huh7-$GCK^+/HK2^-$ cells also resulted in increased secretion of inflammatory interleukins (IL-6 and IL-8), antiviral cytokines (IFN-$\lambda_1$, IFN-$\lambda_{2/3}$, IFN-$\beta$), and IP-10 (Fig. 6e), indicating that both NF-κB- and IRF3-dependent signaling pathways were induced. IL-1β, TNFα, IL-12p70, GM-CSF, IL-10 and IFNγ were not detected in the supernatants of none of the cell lines, whether they were stimulated or not.

As natural killer (NK) cell-mediated lysis of tumor cells is crucial for the anti-cancer immune defense, we compared the susceptibility of the two cell lines to NK cells cytotoxicity. Figure 6f shows that Huh7 cells are resistant to NK cell-mediated lysis in contrast to Huh7-$GCK^+/HK2^-$. Thus, replacing HK2 by GCK restored Huh7 sensitivity to NK cell-mediated lysis. Similar results were obtained when NK cells were pre-activated with IL-2 (Supplementary Fig. 11). Altogether, these results demonstrate that HCC cells expressing HK2 instead of GCK exhibit an impaired response to immune signals and also a strong resistance to NK cells. These two observations are in line with clinical data showing that elevated GCK expression is associated with prolonged survival, while elevated HK2 expression coinciding with GCK reduction correlates with shorter overall survival (Fig. 1).

## Discussion

Metabolic network rewiring is a hallmark of cancer although for most tumors, mechanisms at the origin of this metabolic reprogramming have not been elucidated. While GCK, but not HK2, is expressed in normal hepatocytes, the expression of HK2 occurs during cirrhosis and increases as the disease progresses to carcinoma. Several signaling pathways such as hypoxia inducible

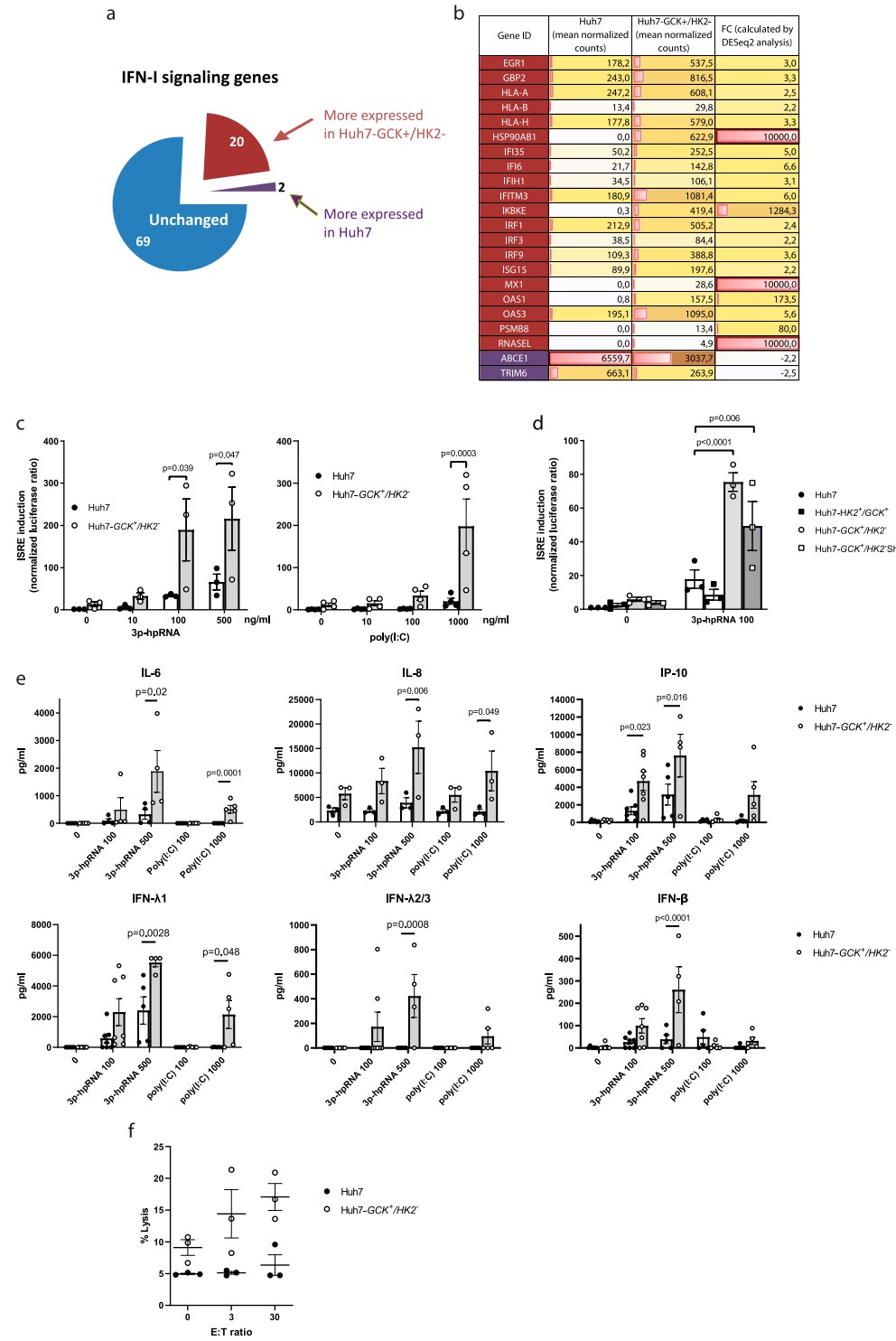

**Fig. 6 Innate immune response is enhanced in Huh7-*GCK$^+$/HK2$^-$* cells. a** Sector chart from the transcriptomic study showing genes included in the GO-term "Type I-IFN signaling pathway". **b** List of genes significantly up-regulated in red or down-regulated in purple (| FC | > 2, BH adjusted $p$ value<0.05) in Huh7-*GCK$^+$/HK2$^-$* compared to Huh7 cells ($n = 3$). **c–e** Cells were stimulated or not for 48 h with 3p-hpRNA (RIG-I ligand) or poly(I:C) (IFIH1/MDA5 ligand). ISRE-luciferase expression was monitored and normalized to Renilla luciferase (**c**, **d**) ($n = 3$ for 3p-hpRNA and $n = 4$ for poly(I:C) treatments). Cell supernatants were assayed for cytokine concentration by multiplex assays ($n = 3$ to 7) (**e**). **f** NK cell mediated lysis of Huh7 or Huh7-*GCK$^+$/HK2$^-$* cells. Hepatoma cells were seeded 24 h before NK cells addition for 4 h at effector to target (E:T) ratio of 0, 3 or 30. After harvesting, cell lysis was determined by the percentage of PI$^+$ cells on gated hepatocytes ($n = 3$). Means ± SEM of indicated $n$ independent experiments are presented and p values were obtained from 2-way ANOVA analyses comparing matched cell means with Sidak's correction for multiple comparison, with α = 0.05.

factors (HIF), peroxisome proliferator-activated receptors (PPAR) and phosphatidylinositol-4,5-bisphosphate 3-kinase (PI3K) might contribute to HK2 induction in fatty liver disease and its evolution towards cirrhosis and carcinogenesis[35–37]. Consequently, HK2 induction has been proposed as a risk marker of HCC development[16]. Analyzing TCGA data from human HCC tumors, we observed that not only high levels of *HK2* but also low levels of *GCK* are of poor prognosis. In contrast, neither *HK1* nor *HK3* expression levels were correlated with survival of HCC patients. *GCK* expression is very low or not detected in biopsies from a majority of patients (65.8% of patients show RSEM values <10), whereas *HK2* is widely expressed[16] (only 5.8% of patients show RSEM values <10). This probably explains that HK2 expression is a better prognostic marker than GCK for HCC. However, when GCK and HK2 expression were combined into a single ratio, this prognostic marker outperformed HK2 or GCK expression alone. This suggests that both HK2 induction and GCK loss play a role in HCC progression. As HK2 and GCK expression tend to be mutually exclusive, both HK2 induction and GCK downregulation might have consequences on the metabolic reprogramming during malignant transformation of hepatocytes. To compare the functional consequences of the HK isoenzyme switch in HCC, we therefore expressed GCK in the reference HCC cell line Huh7 and knocked-down HK2 expression. Our comparative transcriptomic, metabolic and functional studies demonstrate that the replacement of HK2 by GCK not only restored some essential metabolic functions of normal hepatocytes such as lipogenesis, VLDL secretion and glycogen storage but also reactivated innate immune responses and sensitivity to NK-mediated cell lysis.

HCC cell lines predominantly secrete LDL-like particles, unlike normal hepatocytes, which secrete VLDL. Lipid loading of Huh7 cells with oleic acid can boost the secretion of ApoB$^+$ particles but does not induce a shift from LDL to VLDL density, indicating that intracellular fatty acid accumulation of exogenous origin cannot rescue VLDL production[28]. Here we show that replacing HK2 by GCK in Huh7 cells restored de novo fatty acid synthesis, allowing VLDL assembly/secretion in the absence of exogenous lipids. To our knowledge Huh7-*GCK*$^+$/*HK2*$^-$ is the first human cell model with a functional VLDL secretion pathway. Such a tool will strongly benefit the field of cardiovascular diseases and hepatic steatosis.

De novo fatty acid synthesis from carbohydrates requires an adequate supply in metabolic substrates, especially citrate that is produced by the TCA cycle from incoming pyruvate. The glycolytic entry point into the TCA cycle is controlled by PDH and PC that convert pyruvate into acetyl-CoA or OAA, respectively. Our data revealed that in addition to the increased production of pyruvate from glucose, PC activity is increased whereas PDH is inhibited. This suggests that pyruvate metabolism is rebalanced in favor of OAA in Huh7-*GCK*$^+$/*HK2*$^-$ cells, as described in healthy liver. Such a mechanism of anaplerosis is known to replenish TCA cycle intermediates and compensate citrate export out of the mitochondria for lipogenesis fueling. Increased PC activity is observed in both normal and pathological situations, mainly as a result of an increased transcription of the PC gene. In our model, mRNA and protein levels were not affected, indicating that PC activity can be regulated by alternative mechanisms depending on HK isoenzyme expression. This may relate to lower levels of oxalate, a known inhibitor of PC activity, in Huh7-*GCK*$^+$/*HK2*$^-$ cells (Fig. 5c and Fig. 7 discussed below).

A rebalanced pyruvate usage in favor of OAA is also described for instance in SDH-deficient neuroendocrine tumor cells, where succinate accumulates and PC activity is increased to maintain OAA production, replenish the oxidative TCA cycle and support aspartate synthesis[29]. Interestingly, in comparison to Huh7 cells,

succinate and aspartate levels are elevated in Huh7-*GCK*$^+$/*HK2*$^-$ where SDH activity is reduced, suggesting a direct link between PC and SDH activity in hepatocytes. Several mechanisms inhibiting SDH have been described[38]. Modification of the expression of SDH subunits is unlikely as no variation was observed at the transcriptomic level. Itaconate is a weak inhibitor of SDH produced from aconitate by Immune-responsive gene 1 protein (IRG1; encoded by *ACOD1*), but this metabolite was not detected and IRG1 mRNA was absent from the transcriptome of both cell lines. Whether fumarate or other metabolites are responsible for the reduced SDH activity in GCK-expressing cells remains to be investigated. Finally, SDH-deficient cells and LPS-stimulated macrophages have been shown to elicit a hypoxic-like phenotype through accumulation of large amounts of succinate and stabilization of HIF-1α[39,40]. Despite an elevated succinate steady-state level in Huh7-*GCK*$^+$/*HK2*$^-$ compared to Huh7 cells, we observed no difference in HIF-1α stabilization neither at basal level nor upon induction (Supplementary Fig. 12). This suggested that the reduction of SDH activity in Huh7-*GCK*$^+$/*HK2*$^-$ cells was not strong enough to induce such a pseudo-hypoxic phenotype.

Our gene-centric metabolic analysis of transcriptomic data revealed a wide spreading of metabolic modifications resulting from HK isoenzyme switch. Illustrating these modifications, Fig. 7 is an attempt to integrate the observed changes in central carbon metabolism and closely connected metabolic pathways. In particular, decreased level of alanine and increased aspartate concentration in Huh7-*GCK*$^+$/*HK2*$^-$ cells could be an indirect effect of PC activation that uses pyruvate for the synthesis of OAA. As a consequence, hepatic transaminases may balance intracellular pools of OAA, aspartate, alanine and pyruvate. Glutamate and GABA levels were also modified, thus supporting anaplerosis of the TCA cycle through glutamine consumption and the GABA shunt pathway, respectively. We also observed lower levels of oxalate, an end-product of glyoxylate degradation. In Huh7-*GCK*$^+$/*HK2*$^-$ cells, increased levels of alanine-glyoxylate and serine-pyruvate aminotransferase (*AGXT*) could account for this phenotype as it converts alanine and glyoxylate into pyruvate and glycine, which is also increased. Interestingly, high level of *AGXT* is a good prognostic marker for HCC[41]. Consistently, it was found that oxalate inhibits liver PC, resulting in reduced gluconeogenesis and lipogenesis[42,43]. Thus, a higher PC activity could be explained by lower levels of oxalate in Huh7-*GCK*$^+$/*HK2*$^-$ cells. We also observed that isoleucine and valine levels increased while branched chain amino acid transaminase 1 (*BCAT1*) predominant transcripts decreased. This suggests a reduced catabolism of branched chain amino acids in Huh7-*GCK*$^+$/*HK2*$^-$ cells. Again, low levels of *BCAT1* is a good prognostic marker for HCC and oral supplementation with branched chain amino acids has been shown to reduce the risk of liver cancer in cirrhotic patients[44,45]. If some metabolic modifications seem to advocate for the restoration of a normal hepatocyte phenotype following the replacement of HK2 by GCK, it cannot be a general statement. Indeed, the urea cycle was also impacted in Huh7-*GCK*$^+$/*HK2*$^-$ cells with lower levels of *CPS1* and *OTC*, which are also observed in aggressive HCC tumors[46]. Altogether, our results demonstrate the broad impact of replacing HK2 by GCK in HCC cells, and the key role played by the HK isoenzyme switch in HCC tumor metabolism.

We discovered that HK isoenzyme expression not only controls hepatic metabolic functions but also interferes with intrinsic innate immunity of hepatocytes and antitumor immune surveillance. Several reports have recently established functional links between glucose metabolism and signaling pathways downstream of innate immunity sensors of the RLR family, RIG-I and MDA5[47–49], which are usually associated with antiviral responses. However, *RIG-I* expression is downregulated in hepatic cancer

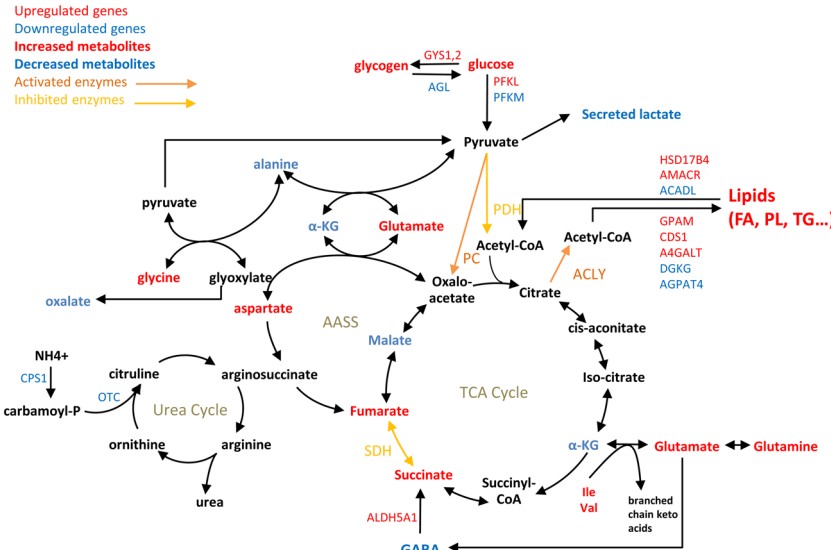

**Fig. 7 Simplified scheme of central carbon metabolism and connected pathways showing differences between Huh7-*GCK*⁺/*HK2*⁻ vs Huh7.** Highlighted metabolites, enzymatic activities, and metabolism-associated genes were selected from transcriptomic (Fig. 2e), metabolomic (Fig. 5c) and enzymatic analyses (Fig. 5f–l).

tissues and low RIG-I expression is correlated with poor survival of patients, whereas *RIG-I* expression in HCC cell lines enhanced IFN response and cancer cell apoptosis[34]. This suggests an unexpected role of this receptor family in the antitumor response. Here we show that Huh7 cells expressing GCK instead of HK2 exhibit a higher sensitivity to RIG-I and MDA5 ligands, and produce higher levels of type I/III IFNs and inflammatory cytokines. This immune phenotype occurs in a context of reduced SDH activity and increased intracellular content in succinate (Fig. 5k, l). A pro-inflammatory function of immune cells such as macrophages was previously linked to TCA rewiring, with reduced SDH activity resulting in succinate accumulation[39,50,51]. Succinate can also be secreted from LPS-activated macrophages and activate its cognate receptor, succinate receptor 1 (SUCNR1, previously known as GPR91) in an autocrine and paracrine manner to further enhance production of IL-1β[52]. Interestingly, glucose metabolism promotes RIG-I and MDA5 signaling through the O-GlcNAcylation of the mitochondrial adaptor MAVS[47]. Thus, an intriguing hypothesis is that GCK expression could facilitate MAVS signaling by increasing UDP-GlcNAc through upregulation of the hexosamine biosynthetic pathway. HK2 binding at the surface of mitochondria may also compete with pyruvate carboxylase, metabolites or mitochondria factors known to control MAVS signaling[47–49]. Here we show that HK2 knockdown promotes RIG-I-induced ISRE-dependent transcription (Fig. 6d). This is consistent with the results obtained by Zhang et al.[48], indicating that HK2 interaction with MAVS restrains RIG-I-induced IFN-β secretion. Further investigations are now required to decipher the molecular links between metabolism and immune responses.

Beyond the inhibition of RLR signaling, other mechanisms might contribute to tumor escape from immune surveillance in HCC patients. In advanced-stage HCC patients, NK cells often exhibit reduced infiltration and impaired functional activities[53]. We thus compared the sensitivity of Huh7-*GCK*⁺/*HK2*⁻ cells to Huh7, and found that sensitivity to NK cell lysis is restored when HK2 is replaced by GCK. When analyzing cell surface expression of the NK cells inhibitors, HLA class I and MICA/B, no significant changes were observed between cell lines (Supplementary Fig. 13). In contrast, an increased transcription and surface expression of ICAM1 (FC = 2.6; Supplementary Fig. 13) was

observed in Huh7-*GCK*⁺/*HK2*⁻. Since ICAM1 binding to active LFA-1 at the surface of NK cells is essential for granule polarization and efficient killing of the target cells[54], its enhanced exposition at the surface of Huh7-*GCK*⁺/*HK2*⁻ cells may contribute to their higher sensitivity to NK cell-mediated killing. These results suggest that HK2 expression at the expense of GCK in HCC tumors decreases immune responsiveness and sensitivity to NK cytotoxicity, thus favoring immune escape.

Taken together, our data demonstrate that beyond glycolysis, the hexokinase isoenzyme switch in an HCC model rewires central carbon metabolism, promotes lipogenesis, enhances innate immune functions, and restores sensitivity to natural killer cells.

## Methods

**Materials.** Unless otherwise specified, chemicals were from Merck Sigma-Aldrich. The RIG-I specific ligand 3p-hpRNA and the MDA5/TLR3 ligand poly(I:C) HMW (High Molecular Weight) were from Invivogen.

**Cell cultures.** Cell cultures were tested negative for mycoplasma contamination by PCR (mycoplasma check, eurofins). Huh7 cells were authenticated by Eurofins Medigenomix Forensik GmbH using PCR-single-locus-technology. 21 independent PCR-systems Amelogenin, D3S1358, D1S1656, D6S1043, D13S317, Penta E, D16S539, D18S51, D2S1338, CSF1PO, Penta D, TH01, vWA, D21S11, D7S820, D5S818, TPOX, D8S1179, D12S391, D19S433 and FGA (Promega, PowerPlex 21 PCR Kit) were investigated to determine their genetic characteristics. Huh7 cells and derivatives were grown as previously described[55] in DMEM, 10% fetal calf serum (FCS), penicillin/streptomycin, 1 mM pyruvate, 2 mM L-glutamine. Culture medium and additives were from Gibco except FCS (Dominique Dutcher).

**Cell lines.** 15×10⁴ Huh7 cells were transduced for GCK expression at different multiplicities of infection (lentiviral transduction using the pLEX-GCK construct). The Huh7-*GCK*⁺/*HK2*⁺ cells were then cultured for 7 days with puromycin (1 μg/mL) before amplification. HK2 knock-out was achieved using the CRISPR/ Cas9 system as previously described[56] to obtain Huh7-*GCK*⁺/*HK2*⁻ cells. Briefly, a single guide RNA (sgRNA) pair was designed for double nicking using the CRISPR Design Tool (http://tools.genome-engineering.org). The guide sequence oligos (sgRNA₁(HK2): 5'-CACCGTGACCACATTGCCGAATGCC-3' and sgRNA₂(HK2): 5'-CACCGTTACCTCGTCTAGTTTAGTC-3') were cloned into a plasmid containing sequences for Cas9 expression and the sgRNA scaffold (pSpCas9(BB)-2A-GFP, Addgene plasmid #48138). 48 h post-transfection, cells were sorted by FACS based on the transient expression of GFP and cloned by limiting dilution. Effective deletion of HK2 was assessed by qPCR.

For HK2 knock-down, Huh7-*GCK*⁺/*HK2*⁺ cells were transduced with lentiviral vectors expressing HK2-targeting shRNAs, and antibiotic selection was applied (hygromycin; 100 μg/ml). The *HK2*-targeting sequence 5'-

CCGGCCAGAAGACATTAGAGCATCTCTCGAGAGATGCTC-
TAATGTCTTCTGGTTTTTT-3' was cloned in the pLKO.1 hygro vector (a gift
from Bob Weinberg; Addgene plasmid #24150). HK2 expression in Huh7-$GCK^+$/
$HK2^+$ and Huh7-$GCK^+$/$HK2^-$Sh was analyzed on cell lysates by western blotting
(Supplementary Fig. 10).

**Enzymatic activity assays**. Cells were trypsinized, washed twice, and cell pellets
were stored at −80 °C. Protein extractions and assays were performed in specific
buffers for hexokinase and pyruvate carboxylase assays as described below.

**Hexokinase activity assay**. The method used for monitoring HK activity in cells
lysates was adapted from Kuang et al.[57–59]. Cellular pellets stored at −80 °C were
thawed and immediately homogenized ($2\times10^6$ cells/100 μl) in precooled reaction
buffer. (0.05 M Tris–HCl, 0.25 M sucrose, 0.005 M EDTA, 0.005 M 2-mercap-
toethanol, pH = 7.4). After 20 min incubation on ice, homogenates were pulse-
sonicated 15 s at half power (EpiShear Probe Sonicator). Homogenates were then
centrifuged at 500 g for 20 min at 4 °C. Supernatants were immediately used for
determination of HK activity, which was measured spectrophotometrically through
$NADP^+$ reduction in the glucose 6-phosphate dehydrogenase–coupled reaction.
HK activity was assayed in medium containing 50 mM triethanolamine (pH = 7.6),
10 mM $MgCl_2$, 1.4 mM $NADP^+$, with variable concentration of glucose and 1 U
glucose 6-phosphate dehydrogenase (S. cerevisiae), equilibrated to 37 °C. The
reaction was started by addition of ATP (final concentration 1.9 mM), and
absorbance was continuously recorded for 30 min at 340 nm (TECAN
Infinite M200).

**Pyruvate carboxylase activity assay**. The method used for quantification of PC
activity was adapted from Payne et al.[60]. Briefly, cells were centrifuged, washed
twice with ice-cold PBS before homogenization in Tris-HCL 100 mM, pH = 8.0
using a Dounce homogenizer. Homogenates were pulse-sonicated 15 s at half
power (EpiShear Probe Sonicator) before centrifugation at 500 g for 5 min.
Supernatants were immediately used for the assay. PC activity was assayed in
medium containing 100 mM Tris-HCl, 50 mM $NaHCO_3$, 5 mM $MgCl_2$, 0.1 mM
Acetyl-CoA, 0.25 mM 6,6′-Dinitro-3,3′-dithiodibenzoic acid (DTNB), 5 mM ATP,
5 mM pyruvate, citrate synthase and cofactors. Reduction of DTNB by the gen-
erated free CoA was measured continuously by Abs at 412 nm and recorded for
30 min (TECAN Infinite M200). The same assay was performed in absence of
pyruvate to subtract background signal.

**Metabolomics profiling**. Cells were seeded at $13\times10^5$ cells per 75 $cm^2$ dishes. After
24 h, supernatant was removed and replaced by fresh culture medium. For quan-
tification of metabolic flux from glucose, culture medium was supplemented with
both [U-$^{13}$C]-glucose (Sigma-Aldrich; 389374-2 G) and unlabeled glucose at a
50:50 ratio (final concentration of 25 mM glucose). After 24 h, cells were harvested,
washed twice with ice-cold PBS and cell pellets were frozen at -80 °C until meta-
bolites extraction. Cell pellets were transferred into a pre-chilled microcentrifuge
tube with 1 mL cold extraction buffer consisting of 50% methanol (A452, Fisher
Scientific) in ultrapure water. Samples were then frozen in liquid nitrogen, thawed,
and placed in a shaking dry bath (Thermo Fisher Scientific, Waltham, MA) set to
1100 rpm for 15 min at 4 °C. After centrifugation for 15 min at 12500 g and 4 °C
(Sorvall, Thermo Fisher Scientific) using a fixed-angle F21-48×1.5 rotor, super-
natants were collected and dried by vacuum centrifugation overnight. Dried
metabolites were derivatized by addition of 20 μL of 2.0% methoxyamine-
hydrochloride in pyridine (MOX, TS-45950, Thermo Fisher Scientific) followed by
incubation during 90 min in shaking dry bath at 30 °C and 1100 rpm. Ninety μL of
N-methyl-N-trimethylsilyltrifluoroacetamide (MSTFA, 701270.201, Macherey-
Nagel) was added, and samples were incubated and shaken at 37 °C for 30 min
before centrifugation for 5 min at 14,000 rpm and 4 °C. Metabolites were separated
in the supernatant were then separated by gas chromatography (GC, TRACE 1310,
Thermo Fisher Scientific) coupled to a triple-quadrupole mass spectrometry system
for analysis (QQQ GCMS, TSQ8000EI, TSQ8140403, Thermo Fisher Scientific),
equipped with a 0.25 mm inner diameter, 0.25 μm film thickness, 30 m length 5%
diphenyl / 95% dimethyl polysiloxane capillary column (OPTIMA 5 MS Accent,
725820.30, Macherey-Nagel) and run under electron ionization at 70 eV. Using
established separation methods[61–63], the GC was programed with an injection
temperature of 250.0 °C and splitless injection volume of 1.0 μL. The GC oven
temperature program started at 50 °C (323.15 K) for 1 min, rising at 10 K/min to
300.0 °C (573.15 K) with a final hold at this temperature for 6 min. The GC flow
rate with helium carrier gas (HE, HE 5.0UHP, Praxair) was 1.2 mL/min. The
transfer line temperature was set at 290.0 °C and ion source temperature at
295.0 °C. A range of 50–600 m/z was scanned with a scan time of 0.25 s.

**Metabolomics data processing**. Metabolites were identified using TraceFinder
(v3.3, Thermo Fisher Scientific) based on libraries of metabolite retention times
and fragmentation patterns (Metaflux, Merced, CA). Identified metabolites were
quantified using the selected ion count peak area for specific mass ions, and
standard curves generated from reference standards run in parallel. Peak intensities
were median normalized. The mean and standard deviation for each quantified

metabolite was calculated for each cell line and treatment condition. A univariate
*t*-test was used to compare treatment conditions for each metabolite and cell line.

**Transcriptome profiling of Huh7 and Huh7-$GCK^+$/$HK2^-$ cell lines**. Tran-
scriptome profiling was performed by next generation sequencing (ProfileXpert,
Lyon, France). Briefly, Total RNA was extracted and purified from cell pellets using
Direct-zol RNA purification kit (Zymo Research). 700 ng of total RNA were
amplified (NextFlex Rapid Directional mRNA-Seq, PerkinElmer) to generate
mRNA-seq libraries. Then, gene expression was analyzed by Next-Generation
Sequencing (NGS) using Illumina NextSeq500. Reads were mapped on the refer-
ence genome Homo sapiens GRCh37/hg19. Raw data were processed using the
DESeq2 pipeline[64] to identify differentially expressed genes. See Supporting
Information and Gene Expression Omnibus database with the accession number
GSE144214 for entire raw data.

**Pathway analysis**. The list of transcripts differentially expressed in Huh7 and
Huh7-$GCK^+$/$HK2^-$ cell lines was analyzed by gene set enrichment analysis (IPA,
Build version: 486617 M, Qiagen) weighted by their corresponding fold change and
*p* value. The fold change cut-off of mean expression for each transcript was set at 2
with an adjusted *p* value<0.05. The list of genes associated with "Type I-IFN
signaling pathway" was defined in the AmiGO 2 database. Expression data of these
genes were retrieved from the transcriptomes of Huh7-$GCK^+$/$HK2^-$ and Huh7, and
correspond for each gene to the most differentially expressed transcript.

**Cell migration assay**. $2\times10^4$ cells were plated in the upper chamber of transwells
(Sarstedt, PET 8.0-μm, TL - 833932800) with DMEM without FCS to allow
migration for 24 h at 37 °C. DMEM with 10% FCS was distributed in each well,
below the chamber. Chambers were gently picked up before a brief PBS rinse and
0.05% crystal violet coloration. The migrated cells were analyzed a Leica M50
microscope using a magnification factor of 20x. The number of cells that have
migrated through the membrane and attached on the underside of the membrane
were counted using the software Image J.

**Intracellular lipid staining**. For fluorescence microscopy staining of intracellular
lipids, cells were seeded and cultured during 48 h before staining with Oil-Red-O.
Cells were fixed 15 min at RT with a 4% formaldehyde solution, washed twice with
water before a 5 min incubation with isopropanol 60%. Isopropanol was then
removed and Oil-Red-O solution (Millipore Sigma-Aldrich) added on cells for
15 min at RT. Cells were then extensively washed with water to remove the
exceeding dye before nucleus counterstaining with NucBlue Fixed Cell Stain
ReadyProbes reagent (ThermoFisher Scientific) and observation with a Nikon
Eclipse Ts2R microscope (x60). For the quantification of intracellular lipid droplets
by flow-cytometry, cells were stained with the BODIPY® 493/503 dye (Tocris Bio-
Techne) after 48 h of culture. The cells were washed with PBS before being incu-
bated for 5 min with a 5 μM BODIPY solution in PBS at 37 °C. Cells were then
washed with PBS before trypsination and FACS analysis. A 7-AAD (BioLegend)
staining of dead cells, prior to FACS analysis, allowed gating on living cells.

**Protein, ApoB, and lipid quantification**. Protein concentration was determined
using the DC Protein Assay (Bio-Rad). ApoB concentration in medium and gra-
dients fractions was determined by ELISA as previously described[65]. Total con-
centrations of cholesterol, phospholipids, and triglycerides (TG) were determined
using specific assays from Millipore Sigma-Aldrich (ref. MAK043, MAK122 and
MAK266 respectively). Free Fatty Acids were quantified using a specific assay kit
from Abcam (ref. ab65341).

**Iodixanol density gradients**. Iodixanol gradients were prepared as previously
described[66]. One ml of culture supernatant was applied to the top of 6 to 56%
iodixanol gradients and centrifuged for 10 h at 41,000 rpm and 4 °C in a SW41
rotor. The gradient was harvested by tube puncture from the bottom and collected
into 22 fractions (0.5 ml each). The density of each fraction was determined by
weighing.

**Metabolic network coherence computational analysis**. In order to measure the
consistency of differentially expressed genes with a metabolic network, we
employed the metabolic network coherence measure introduced by Sonnenschein
et al.[67]. This approach was previously applied to various disease-related tran-
scriptome profiles[68,69] and for extracting information on the genetic control of
metabolic organization[70]. Recently, detailed theoretical analysis of the extended
version of the method has been performed by Nyczka and Hütt[71]. Here, we first
extracted a gene-centric metabolic network from a given genome-scale metabolic
model. This was achieved via the stoichiometric matrix and the gene-reaction
associations contained in the metabolic model. We constructed the two projections
of the bipartite graph represented by the stoichiometric matrix, yielding a
metabolite-centric and a reaction-centric graph. The metabolite-centric graph
allowed us to identify high-degree nodes ('currency metabolites' like $H_2O$, ATP,
etc.), which are not informative about the network-like organization of the meta-
bolic systems and need to be eliminated before interpreting the network

architecture (see references[68,72] for details). The degree of a node is the number of neighbors the node has in the network. The percentage of remaining metabolites is one of the parameters of our analysis. Typical values are 90 to 98 percent (i.e. a removal of the highest 2 to 10% of metabolites with the highest degree as currency metabolites). After recomputing the reaction-centric graph based on the reduced number of metabolites (Supplementary Fig. 3), we can now evaluate the gene-reaction associations to arrive at a gene-centric metabolic network (Supplementary Fig. 3). Given a set S of differentially expressed genes and the gene-centric metabolic network G, we can now analyze the subgraph of G spanned by all genes in S. The average clustering coefficient C in these subgraphs serves as a measure of the connectivity of this subgraph. The metabolic network coherence MC is the z-score of C computed with respect to a null model of randomly drawn gene sets with the same size as S (Supplementary Fig. 4). In this way, MC has an intuitive interpretation: The value of MC indicates, how many standard deviations away from randomness the clustering of the subgraph spanned by the observed gene set S actually is (Supplementary Fig. 4 and reference[73]). The genome-scale metabolic models employed here are the generic human metabolic model Recon 2[22]. In general, different network measures can be used for evaluation of MC. In the scope of this study, we have tested several of them, but opted for average clustering coefficient C, as it yielded strongest statistical signal.

**Western-blot analysis**. Cell lysates from $10^6$ cells were prepared in lysis buffer (1% Triton X-100, 5 mM EDTA in PBS with 1% protease inhibitor cocktail (P8340; Millipore Sigma-Aldrich) and 2 mM orthovanadate). After elimination of insoluble material, proteins were quantified, separated by SDS-PAGE and analyzed by western-blot on PVDF membrane. After saturation of the PVDF membrane in PBS-0.1% Tween 20 supplemented with 5% (w/v) non-fat milk powder, blots were incubated 1 h at room temperature with primary antibody in PBS-0.1% Tween 20 (1:2,000 dilution for all antibodies unless specified otherwise). Incubation with secondary antibody was performed after washing for 1 h at room temperature. HRP-labeled anti-goat (Santa Cruz Biotechnology), anti-rabbit (A0545, Millipore Sigma-Aldrich) or anti-mouse (Jackson ImmunoResearch Laboratories) antibodies were diluted 20,000 folds and detected by enhanced chemiluminescence reagents according to the manufacturer's instructions (SuperSignal Chemiluminescent Substrate, Thermo Fisher Scientific). Primary antibodies used for immunoblotting included mouse monoclonal antibody against human GCK (clone G-6, Santa Cruz Biotechnology), rabbit monoclonal antibody against human HK2 (Clone C64G5, Cell Signaling Technology), rabbit monoclonal antibody against human HK1 (C35C4, Cell Signaling), rabbit polyclonal antibody against human HK3 (HPA056743, Millipore Sigma-Aldrich), goat polyclonal antibody against human ACLY (SAB2500845, Millipore Sigma-Aldrich), rabbit polyclonal antibody against human pACLY (phospho S455, Cell Signaling Technology), rabbit monoclonal antibody against human PDH α1 subunit (C54G1, Cell Signaling Technology), rabbit monoclonal antibody against human pPDH E1-alpha subunit (phospho S293, Abcam), goat polyclonal antibody against human PC (SAB2500845, Millipore Sigma-Aldrich), rabbit monoclonal antibody against human GAPDH (D16H11, Cell Signaling Technology) and rabbit polyclonal antibody against human HIF-1α (NB100-134, Novus Biologicals; 1:500 dilution).

**Respiration assay**. Twenty-four hours prior measuring respiration in the Extra-cellular Flux Analyzer (Seahorse Bioscience), cells were seeded in XF 24-wells cell culture microplates (Seahorse Bioscience) at $5×10^4$ cells/well in 100 µL of DMEM medium supplemented with 10% FCS, 1 mM pyruvate, 2 mM L-glutamine, penicillin/streptomycin, and then incubated at 37 °C/5% $CO_2$ during 5 h for cell attachment. Medium volume was adjusted to 250 µL and cells incubated overnight. The assay was initiated by removing the growth medium from each well and replacing it with 500 µL of Seahorse assay medium (XF DMEM pH = 7.4 + 10 mM Glucose, 2 mM Glutamine and 1 mM sodium pyruvate) prewarmed at 37 °C. Cells were incubated at 37 °C for 1 h to allow media temperature and pH to reach equilibrium before the first measurement. The oxygen consumption rate (OCR) was measured using the following Seahorse running program: injection Port A – 1.5 µM Oligomycin; Injection Port B – 0.5 µM FCCP and injection Port C – 0.5 µM Rotenone; injection port D – 0.5 µM Antimycin A. The number of cells was determined at the end of the run after Hoechst staining and cell counting using Cytation 1 imaging reader (Biotek).

**RLR stimulation**. Cells were seeded in 96-well or 24-well plates. After 24 h, cells were co-transfected with indicated doses of the RIG-I ligand 3p-hpRNA or the MDA5/TLR3 ligand poly(I:C) HMW together with the pISRE-luc (1.25 µg/ml; Stratagene) and pRL-SV40 (0.125 µg/ml; Promega) reporter plasmids using the JetPEI-Hepatocyte reagent (Polyplus Transfection). Manufacturer's instructions were followed. After 48 h, supernatants were collected for cytokine quantification. Firefly and Renilla luciferase expressions within cells were determined using the Dual-Glo luciferase Assay system (Promega) and an Infinite M200 microplate reader (TECAN).

**Cytokine assays**. Clarified culture supernatants were collected and stored at −20 °C. IL-8 was quantified using the Cytometric Bead Array for human IL-8 (BD Biosciences). Other cytokines were assayed using the LEGENDplex multiplex assay (Human Anti-Virus Response Panel, BioLegend). Fluorescence was analyzed using a FACS Canto II (BD Biosciences).

**Human NK cell purification**. NK cells were isolated from human buffy coats of healthy donors obtained from the Etablissement Français du Sang. Informed consent was obtained from donors and experimental procedures were approved by the local institutional review committee. PBMCs were isolated by standard density gradient centrifugation on Ficoll-Hypaque (Eurobio). Mononuclear cells were separated from peripheral blood lymphocytes (PBLs) by centrifugation on a 50% Percoll solution (GE Healthcare). NK cells were purified from PBLs by immuno-magnetic depletion using pan-mouse IgG Dynabeads (Thermo Fisher Scientific) with a cocktail of depleting monoclonal antibodies: anti-CD19 (4G7 hybridoma), anti-CD3 (OKT3 hybridoma, ATCC, Manassas, VA, USA), anti-CD4, anti-CD14 and anti-glycophorin A (all from Beckman Coulter). NK purity was >70% as assessed by CD56 labeling.

**NK cell cytotoxicity test**. Huh7 or Huh7-$GCK^+/HK2^-$ were seeded at $1×10^5$ cells per well in a 24-well plate in RPMI-1640 (Gibco) with 10% FCS and 40 µg/ml gentamycin. After 24 h, $3×10^5$ or $3×10^6$ NK cells were added to the culture wells. The cytotoxicity assay was performed for 4 h at 37 °C, under 5% $CO_2$. Target hepatoma cells were harvested after trypsination, labeled with propidium iodide (PI) and analyzed by FACS. Cell death was monitored after morphological gating on hepatocytes.

**Statistics and reproducibility**. All the statistical analyses were performed with GraphPad Prism or Analyse-it software. Details of statistical analyses can be found in figure legends. Two-sided statistical analyses were performed on experiments reproduced at least 3 times independently. The exact p values are indicated either directly in the figure or in the legend. The exact sample size (n) is given in the legend of each figure. The mean ± standard error of the mean (SEM) is displayed, unless otherwise stated. Confidence interval was set to 95% in all statistical tests.

**Reporting summary**. Further information on research design is available in the Nature Research Reporting Summary linked to this article.

## Data availability
The data generated or analyzed during this study are included in the article and supplementary files. The transcriptomes of the 365 HCC biopsies analyzed in the current study were obtained from The Cancer Genome Atlas (TCGA) database and are available in Supplementary Data 1. The RNA-seq data for Huh7 and Huh7-$GCK^+/HK2^-$ cell lines are available in Supplementary Data 2 and at the Gene Expression Omnibus database with the accession number GSE144214 for entire raw data. Source data and calculations for all experiments can be found in Supplementary Data 4. Uncropped images of western are provided in Supplementary Figs. 1 and 7.

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

## Acknowledgements

We acknowledge the contribution of the Genomics and Microgenomics platform ProfileXpert (University Lyon 1, SFR santé LYON-EST, UCBL-Inserm US 7-CNRS UMS3453) and SFR Biosciences (UMS3444/CNRS, US8/Inserm, ENS de Lyon, UCBL) facilities: AniRA-Cytometry, AniRA-ImmOs metabolic phenotyping and LYMIC-PLATIM microscopy. We gratefully thank Laurence Canaple for technical assistance. This work was supported by the Fondation pour la Recherche Médicale (FRM), grant number DEQ20160334893 to VL. F.V.F. is grateful for the support by grants CA154887, GM115293, CRN-17-427258, NSF GRFP, and the Science Alliance on Precision Medicine and Cancer Prevention by the German Federal Foreign Office, implemented by the Goethe-Institute, Washington, DC, USA, and supported by the Federation of German Industries (BDI), Berlin, Germany.

## Author contributions

L.P.-C., P-O.V., O.D. and V.L. designed the experiments with critical advices from G.J.P.R., P.A., R.R. and F.V.F.; C.J., A.A.-G., K.O., B.P., G.J.P.R., N.A, R.R. and F.V.F. performed experiments and analyzed the data; P.N. and M-T.H. performed metabolic network computational analysis; L.P.-C., P-O.V., P.A., F.V.F, V.L. and O.D. analyzed the data, prepared figures and wrote the manuscript.

## Competing interests

The authors declare no competing interests.
