## [Peer Review File · Communications Biology]

Reviewers' comments:

Reviewer #1 (Remarks to the Author):

The manuscript submitted by Perrin-Cocon et al is an original piece of experiments aimed at investigating the impact of switching the phosphorylase capacity of the Huh7 HCC cell line between Hexokinase 2 and Glucokinase, the latter being expressed preferentially in normal hepatocytes. The hypothesis underlying their work is that the activity/expression of glucokinase is essential for the adoption of a normal hepatocellular phenotype away from the carcinogenic characteristics of the original Huh7 cell line. The authors report that the expression of glucokinase together with hexokinase-2 repression changes the metabolism of the cell line that regains the capacity to form and export fatty acids and that shows a more robust response to of innate immunity signals.

Main critique:

The rationale and the hypothesis are sound. The authors have used adequate models to demonstrate their hypothesis. Results are convincing and should be easily reproducible. The conclusion is generally supported by the data.

Minor comments:

The difference in overall survival between the groups (Figure 1A) is not that striking although statistically significant. Other sub-groups could have been chosen (different degrees of expression) or alternative explanations discussed to explain in particular, the limited impact of GCK expression. The in vitro behaviour of the 2 cell lines should be addressed: how different are the 2 cell lines with regards to proliferation, sensitivity to chemo agents, degree of invasiveness? Testing motility is not enough, especially if the authors want to conclude that their results mean that the expression of glucokinase is important for differentiation.

The inversed correlation between glucokinase and hexokinase-2 needs to be better defined: the r coefficient (Figure 1B) needs to be added.

The conclusion written in the abstract ("playing a role in dedifferentiation") is different than the once stated in their final conclusion (play a role in the metabolic and immune status of hepatocytes during malignant transformation).

Have the authors measured glucose 6-PO4 levels in the 2 respective cell lines?

I am surprised by the absence of comments on the increased lactate levels found in the Huh7-GCK+/HK2- cells: this needs to be addressed.

I question the gene centric metabolic analysis of transcriptomic data: I cannot comment on the methods since I do not have the expertise. The authors make the point that there are modifications of gene expression among connected clusters: however, the changes in gene expression are sometimes in opposition inside a given cluster (vg mevalonate). I feel this information might be contradictory. This part of the data does not add meaningful evidence to me and I would tend to remove it from the results.

The last part of the result section is very adequate but is restricted to innate immunity. I propose that this be written in the heading and in the figure legends.

In the discussion, comments on the mechanisms by which Hk2 expression increases in cirrhosis would be welcome.

Since citrate is an essential substrate of lipogenic pathway, have the authors explored the possibility that citrate be provided by the cell culture medium?

The authors write that PC activity can be regulated by alternative mechanisms depending on HK isoenzyme expression: can this be further detailed?

Reviewer #2 (Remarks to the Author):

General Comments:

During development of hepatocellular carcinoma cancer cells undergo a metabolic switch by suppressing glucokinase and elevating hexokinase 2. This study focuses on the consequences of glucokinase-to-hexokinase2 switch by replacing HK2 with GCK in the HCC cell line Huh7. The authors report that the GCK-to-HK2 isoenzyme switch remodel the central carbon metabolism and restored some metabolic and immunological features of hepatocytes. The topic is interesting, however the data presented are mostly preliminary in nature and require substantial improvements. This study is heavily affected by technical and conceptual problems as outlined below.

Major:

- 1: The authors write in the abstract that ' a switch from GCK to HK2 is occurring during the transition from primary hepatocytes to hepatocellular carcinoma. It is therefore counterintuitive that the authors find that 250 HCC tumor display high expression of GCK while only 115 have low expression. Further, although the spearman correlation test was performed to assess the correlation between HK2 and GCK expression, the correlation coefficient is not indicated. For the analysis in Fig1 to be any meaningful, it is critical to compare the same patients displaying either high and or low expression of the studied enzymes. The optimal stratification method used in Fig 1 does not add much and leads to incorrect conclusion.
- 2: The authors generate a HK2 knockout Huh7 cell line which is then replaced by GCK. How is this possible if HK2 is required to sustain proliferation of HCC cells (as stated by the authors in the abstract)? Do Huh7 cells proliferate faster than the GCK-HK2 line? If so, the data needs to be corrected and normalized for this. Further, only the Huh7 cell line is included in this study. Confidence would be significantly increased by including both primary hepatic cells as well as a cell line panel in the analysis. The HK1 and HK3 needs to be added to in Fig2A and it is unclear why HK activity was measured from frozen and not fresh material? Several controls are missing in Fig 2B, for instance knockdown of HK1 and HK3, as well as does re-introduction of GCK affect the expression of the other hexokinase 1-3?
- 3: The Huh7-GCK+/HK2- cells showed a higher migratory capacity with lipid droplet accumulation. As cell migration plays an important role in tumor metastasis. Lipid accumulation is also increased in different neoplastic processes with undergoing cell proliferation. Again, it would be required to assess the cell proliferation. Secondly, this seems to be in conflict with the conflict to the Kaplan-Meier results. Needs clarification.
- 4: For next generation sequencing data analysis, the Student's t-test with Benjamini -Hochberg correction was used to find the differentially expression genes. However, considering that the distribution of the data is not that expected by a Student's t-test. A more appropriate method should be used incl. EdgeR or DESeq2 on the raw counts to identify the correctly identify differentially expressed genes.
- 5: One important function of hepatocytes is to secrete triglyceride-rich VLDL. In contrast, the HCC cells produce less VLDL. Again, the data of the primary hepatic cells should be included to have a proper control.
- 6: To study a snap shot of the metabolites of one cell line with or without genetic modifications does not add much. A proper isotopomer flux analysis needs to be performed to resolve the metabolic reprogramming. It is also surprising that the authors find that the Huh7-GCK+/HK2- cells have modified TCA-cycle and produce more intracellular metabolites compared to Huh7 cells. Cancer cells (which is here more represented in Huh7 cell line) are expected to produce more metabolites to support their growth. Again, this can easily be resolved by performing isotopomer analysis. Although Huh7-GCK+/HK2- cells have higher sensitivity to RIG-I and MDA5 ligands, the authors completely lack to provide any insights into the connection from the rewired metabolism rewiring, ISRE induction, and the role of GCK in prolonged survival. This needs to be resolved.
- 7: The transcriptomic data with the GO term "Type I IFN signaling pathway" was analyzed, however, instead of mentioning just one GO term, the authors should show the results from GO enrichment analysis with significance evaluation and explain why focus on this GO term.
- 8: It is surprising that succinate accumulates in the Huh7-GCK+/HK2- and simultaneously have high respiration. Is Succinate dehydrogenase mutated in the cell line? Authors propose that ' other pathways maintain a high activity of the respiratory chain by fueling complex III' but this was

never tested. It is necessary to repeat the Seahorse data and only provide cells with substrates that feed into complex 1-3, respectively and re-do the assay in order to make such a conclusion.

Minor:

8: Please define the R value in Figure 1B

9: Please define the P value in Figure 4G

10: Specify adjusted p value instead of p value

11: Revise the scientific notation in the method part (e.g. 1.10^5 cells per well, 3.10^5 or 3.10^6 NK cells etc.)

12. Many critical experimental details are missing, for instance which plates (company, cat no) were used for the migration assay? How were the pictures taken? Microscope? If so, objective? Etc. etc.

Manuscript number COMMSBIO-20-1166-T

Response to the reviewers:

Reviewer #1:

Minor comment 1

The difference in overall survival between the groups (Figure 1A) is not that striking although statistically significant. Other sub-groups could have been chosen (different degrees of expression) or alternative explanations discussed to explain in particular, the limited impact of GCK expression.

Reply

In Fig. 1a, the patient cohort was stratified into two subgroups based on individual expression levels of HK. For each HK, the subgroups with the lowest p-value when analyzing survival outcome are presented. This makes the stratification in Fig. 1a optimal for each HK as explained in Uhlen M. & al. (Uhlen M. et al., 2017). Therefore, other subgroups would be less relevant. Clarification was inserted in the main text lines 103-105, "For each HK, the individual gene expression level was used to stratify patients into two subgroups according to Uhlen et al. (18) and overall survival in the two subgroups was determined using a Kaplan-Meier's estimator."

We agree that HK2 expression is more discriminant than GCK expression, but both represent statistically significant prognostic markers. To further support this statement, we introduced a supplementary panel in the Figure 1 (Fig. 1b) showing that the GCK/HK2 ratio outperforms individual markers to stratify patients. Indeed, when patients were stratified on the basis of HK2 or GCK expression, the median survival between the corresponding subgroups differed by 33.8 and 36.5 months, respectively (Fig. 1a). This difference reached 42.8 months when the stratification of patients was based on the GCK/HK2 ratio (Fig. 1b).

In addition to new Fig. 1b, modifications appear as follow in the revised manuscript - lines 108-114, "We thus stratified patients based on the GCK/HK2 expression ratio to combine these two markers (Fig. 1b). When patients were stratified on the basis of HK2 or GCK expression levels, the median survival between the corresponding subgroups differed by 33.8 and 36.5 months, respectively (Fig. 1a). This difference reached 42.8 months when the stratification of patients was based on the GCK/HK2 ratio (Fig. 1b). This demonstrated that the GCK/HK2 ratio outperforms HK2 or GCK expression alone as predictor of patient survival."

Finally, we discuss why GCK expression is not as good as HK2 expression for patients' stratification and overall survival prediction. In total, 240 out of the 365 patients' samples show no or very low GCK expression levels (RSEM value <10, Supplementary Table 1) compared to only 21 patients' samples exhibiting no or very low HK2 expression levels. Thus, GCK expression is lost in a majority of patients, whereas HK2 is induced even before the onset of HCC (Lee N.C.W. et al., 2018) and its expression greatly differs from patient to patient. This would explain why HK2 expression is a better prognostic marker than GCK in the cohort of HCC patients.

This now appears in the discussion section - lines 302-307, "GCK expression is very low or not detected in biopsies from a majority of patients (65.8% of patients show RSEM values <10), whereas HK2 is widely expressed (16) (only 5.8% of patients show RSEM values <10). This probably explains that HK2 expression is a better prognostic marker than GCK for HCC. However, when GCK and HK2 expression were combined into a single ratio, this prognostic marker outperformed HK2 or GCK expression alone. This suggests that both HK2 induction and GCK loss play a role in HCC progression."

Figure 1. Correlation between hexokinase expression levels in HCC tumors and patient survival. **a** Kaplan–Meier estimates of the survival of HCC patients depending on the expression of HK1, HK2, HK3 and GCK (HK4) genes in tumor biopsies ($n=365$; TCGA expression data retrieved from cBioPortal) (70,71). Duplicate analyses from the same patient were removed as well as patients who died when biopsied (overall survival=0 months). Optimal stratification based on highest and lowest gene expression values was determined using Protein Atlas database (18). **b** Same as above but patients were stratified based on the GCK/HK2 gene expression ratio. The stratification showing the lowest p value when comparing subgroups of patients with the highest to the lowest GCK/HK2 expression ratio is displayed. Patient TCGA-DD-AAE9 exhibiting undetectable levels of GCK and HK2 was removed from this analysis as the GCK/HK2 ratio could not be calculated. **c** Correlations between patient survival, GCK expression and HK2 expression. Spearman’s rank correlation test on the subset 130 patients for whom the period between diagnosis and death is precisely known (uncensored data).

Minor comment 2

The in vitro behaviour of the 2 cell lines should be addressed: how different are the 2 cell lines with regards to proliferation, sensitivity to chemo agents? degree of invasiveness? Testing motility is not enough, especially if the authors want to conclude that their results mean that the expression of glucokinase is important for differentiation.

Reply

Cell counts were recorded at each passage of Huh7 and Huh7-GCK⁺/HK2⁻ cell cultures over the last 12 months and were used to calculate doubling time. Over 75 passages, the average doubling-time was not statistically different between the two cell lines (31.8±1.0 and 33.3±1.1 hours for Huh7 and Huh7-GCK⁺/HK2⁻ respectively). We also monitored their proliferation kinetic over 72h of culture and found no difference at any time (see Supplementary Fig. 1). This was mentioned in the revised version of the manuscript - line 132-133, “The cell proliferation capacity remained identical between the two cell lines (Supplementary Fig. 1).”

Supplementary Figure 1: Proliferation of Huh7 and Huh7-GCK⁺/HK2⁻. Cells were seeded in 24 well-plate under standard growth conditions and cellular proliferation was determined at time 0, 24, 48 and 72h post-seeding using the CellTiter-Glo[®] Luminescent Cell Viability Assay (Promega). Luminescence were quantified with an Infinite M200 microplate reader (TECAN). Means ± SEM are presented (n=3).

At this stage, we have not observed major differences in term of chemo-sensitivity but the data are too preliminary to draw conclusions and the question should be addressed in further studies.

The motility test was performed with the dual objective to validate the Ingenuity analysis (functional annotation “cellular movement”) and the functionality of the non-enzymatic role of GCK in regulating cell migration (Beilstein F. et al., 2017) so that both enzymatic and non-enzymatic functions of GCK could be demonstrated in Huh7-GCK⁺/HK2⁻ cells. We do not argue in favor of a differential degree of invasiveness between the two cell lines, which would indeed require a proper in vitro and in vivo investigation. The main message of this paper is that GCK re-expression modulates metabolic and innate immune functions. To avoid over-interpretation of our work, the term “dedifferentiation” is absent from the new abstract.

Minor Comment 3

The inversed correlation between glucokinase and hexokinase-2 needs to be better defined: the r coefficient (Figure 1B) needs to be added.

Reply

In the first version of the manuscript, a dot-graph showed the inverse correlation between GCK and HK2 expression for all patients with available transcriptome data (n=365). Spearman's analysis was performed and calculated r and p-value were -0.122 and 0.019, respectively. Therefore, GCK and HK2 expression levels are inversely correlated at least to a statistical point of view. To better characterize the consequence of GCK and HK2 expression on survival, we calculated the survival correlation coefficient related to their expression on the subset of 130 patients for whom the period between diagnosis and death is precisely known (uncensored data). The Spearman's coefficient between overall survival and HKs expression level is positive for GCK expression but negative for HK2. In addition, we confirmed that GCK and HK2 expression are inversely correlated in this group of 130 patients, and we provide associated r and p-values in the new Fig. 1c. This analysis is discussed in lines 114-119. "Finally, correlation coefficients between patient survival in months and HK2 or GCK expression level were determined. For this, we only considered the subset of 130 patients for whom the period between diagnosis and death is precisely known (uncensored data), and performed a Spearman's rank correlation test (Fig. 1c). Patient survival was positively correlated to GCK expression but inversely correlated to HK2 expression in line with the Kaplan-Meier analysis. In addition, GCK and HK2 expression tends to be inversely correlated in tumor samples (Fig. 1c)."

Minor comment 4

The conclusion written in the abstract ("playing a role in dedifferentiation") is different than the once stated in their final conclusion (play a role in the metabolic and immune status of hepatocytes during malignant transformation).

Reply

This was indeed misleading and was corrected in the new abstract.

Minor comment 5

Have the authors measured glucose 6-PO4 levels in the 2 respective cell lines?

Reply

We measured intracellular glucose-6-PO4 in both cell lines and observed a lower concentration in Huh7-GCK⁺/HK2⁻ cells (Figure R1 below). As glucose-6-PO4 quantification is a steady-state measure, we also measured the glycolytic flux using ¹³C-labelled glucose. We found that production of ¹³C-labelled pyruvate from glucose is higher in Huh7-GCK⁺/HK2⁻ cells (data presented in the revised manuscript Fig. 5d-f and see reply to reviewer 2 comment 6), which indicates that glycolytic rate is higher in these cells. This is now shown in the result section paragraph "Differential activity of the tricarboxylic acid cycle (TCA) in Huh7 and Huh7-GCK⁺/HK2⁻".

Figure R1: Huh7 and Huh7-GCK⁺/HK2⁻ cells were harvested, washed with PBS before homogenization by sonication (Active Motif Q120AM at 50% power, 5sec ON, 5sec OFF, 6 pulses). Homogenates were then centrifuged at 13000g for 10min at 4°C. Proteins were removed from supernatant using spin filter 10kDa (millipore Ultracell 10K Amicon ultra ref UFC501096) and Glucose-6-P concentration determined using Glucose-6-Phosphate Assay Kit (Sigma-Aldrich MAK014) according to the manufacturer instructions. Means \pm SEM are presented (student's t test, n=3).

Figure 5. TCA rewiring after hexokinase isoenzyme switch in Huh7 cells. a Glycogen quantification. **b** Creatinine and creatinine-P quantification. **c** This bubble chart compares intracellular metabolomes of Huh7 and Huh7-GCK⁺/HK2⁻ cells.

Metabolite pool sizes larger in Huh7 are indicated in blue, whereas the one larger in Huh7-GCK⁺/HK2⁻ are shown in red. The size of bubbles inversely scales with p values between 5.10^{-2} and 1.10^{-17} of differential metabolomics responses. **d-e** Metabolic fluxes for overall glucose consumption **d** and lactate secretion **e** by Huh7 and Huh7-GCK⁺/HK2⁻ cells. Indicated values correspond to differences in glucose or lactate concentrations in extracellular culture medium before and after 24h of culture. **f** Mass isotopomer distribution vector of pyruvate in cells cultured with [U-¹³C]-glucose. **g** Pyruvate carboxylase (PC) activity determined in cell homogenates. **h** Western-blot analysis of PC expression in Huh7 and Huh7-GCK⁺/HK2⁻ cells. **i** Western-blot analysis of pyruvate dehydrogenase (PDH) E1-alpha subunit phosphorylation at Ser293. **j** RNAseq quantification of pyruvate dehydrogenase kinase 2 (PDK2) and pyruvate dehydrogenase phosphatase 2 (PDP2)(BH adjusted p value<0.05 from transcriptomic data). **k** Western-blot analysis of ATP-citrate Lyase (ACLY) phosphorylation at Ser455. **l** Succinate quantification in cell homogenates. **m** Succinate dehydrogenase (SDH) activity determined in cell homogenates. **n** Oxygen consumption rate (OCR) in Huh7 and Huh7-GCK⁺/HK2⁻ cells was determined with a Seahorse analyzer before and after the addition of oligomycin (Complex V inhibitor), FCCP (uncoupling agent), rotenone (Complex I inhibitor) and antimycin A (Complex III inhibitor). **o** Non-mitochondrial, complex I-dependent and complex III-dependent maximal OCR were calculated from **n**. Data correspond to means ± SEM (n≥3).

Minor comment 6

I am surprised by the absence of comments on the increased lactate levels found in the Huh7-GCK⁺/HK2⁻ cells: this needs to be addressed.

Reply

As aforementioned, we present fluxomic data in the revised manuscript showing a higher consumption of glucose leading to a greater flow of carbon towards pyruvate in Huh7-GCK⁺/HK2⁻ cells (Fig. 5f). This is accompanied by a lower secretion rate of lactate, as now shown in Fig. 5e, suggesting that glucose-derived pyruvate essentially fuels the TCA cycle. These new data are now presented in the revised Fig. 5 and discussed accordingly lines 205-210, "This led to investigate glucose catabolism in further details. Glucose consumption and stable isotope incorporation from [U-¹³C]-glucose into pyruvate were both increased in Huh7-GCK⁺/HK2⁻ cells compared to Huh7 cells (Fig. 5d and f). This increased glycolytic flux together with a reduced lactate secretion (Fig. 5e) is likely to account for the elevation of lactate levels and suggest that the increased pyruvate production essentially fuels mitochondrial TCA cycle in Huh7-GCK⁺/HK2⁻ cells."

Minor comment 7

I question the gene centric metabolic analysis of transcriptomic data: I cannot comment on the methods since I do not have the expertise. The authors make the point that there are modifications of gene expression among connected clusters: however, the changes in gene expression are sometimes in opposition inside a given cluster (vg mevalonate). I feel this information might be contradictory. This part of the data does not add meaningful evidence to me and I would tend to remove it from the results.

Reply

The other reviewer asked to re-analyze the NGS data using DESeq2 protocol. We thus reconsidered the gene centric metabolic analysis using the DESeq2 analyzed transcriptomes. The intersection between the Recon2 metabolic network and differentially expressed genes in Huh7 vs Huh7-GCK⁺/HK2⁻ cells is now presented in the revised Figure 2 (Fig. 2d and e). The main interest of this analysis is to show that metabolic genes that are differentially expressed between Huh7 and Huh7-GCK⁺/HK2⁻ cells preferentially cluster into metabolite-connected modules or pathways. This is illustrated in Fig. 2d presenting the z-score for network clustering when varying the currency metabolite rate and the threshold for differential gene expression. The highest and most significant clustering coefficient was obtained for 98% of currency metabolites and a $\log_2(|FC|) > 3$ for differential gene

expression. The corresponding network is presented in Fig. 2e, showing connections between glycolysis and distant metabolic enzymes involved in GABA shunt, urea cycle, glycogen and lipid metabolism (Fig. 2e). The fact that some genes are upregulated whereas others are downregulated likely reflects feedback regulations to compensate increased or decreased levels in connected enzymes and/or metabolites. This new analysis is presented in lines 141-156, “We first determined the metabolic consequences of the HK isoenzyme switch by mapping the differentially expressed genes onto the a well-established bipartite metabolic network Recon2, connecting gene products and metabolites (Supplementary Fig. 2-4) (22, 23). After trimming highest-degree metabolites as currency metabolites, clusters of genes that are both differentially expressed and connected by common metabolites emerged. Interestingly, we found that across a wide range of analysis parameters, including varying rates of currency metabolites and gene expression fold-change, the differentially expressed metabolic genes are substantially better connected than expected by chance (Fig. 2d). This highlights the specificity of the transcriptomic changes with respect to metabolic pathways. The spanned network presented in Figure 2e corresponds to a stringent fold-change threshold for transcriptomic data ($\log_2(|FC|) > 3$) while removing 2 percent of highest-degree currency metabolites. This network shows connected components within glycolysis, but also across distant modules including the gamma-aminobutyric acid (GABA) shunt (ALDH5A1), urea cycle (CPS1, OTC), glycogen metabolism (GYS1, GYS2, AGL) and lipid synthesis (GPAM, AGPAT4, DGKG, CDS1, A4GALT) or degradation (ACADL, HSD17B4, AMACR). This analysis highlights the global impact of the HK isoenzyme switch that spreads beyond glycolysis across distant connected metabolic modules.”

Figure 2. Hexokinase isoenzyme switch in Huh7 cells induces extended modifications of metabolic connections. **a** Western-blot analysis of HK1, HK2, HK3 and GCK expression in Huh7 and Huh7-GCK⁺/HK2⁻. **b** Hexokinase activity in homogenates of

Huh7 and Huh7-GCK⁺/HK2⁻ cells. c Number of genes changing their expression pattern in Huh7 and Huh7-GCK⁺/HK2⁻ cells (see Supplementary Table 2 for details). d Heatmap showing clustering enrichment scores of the networks obtained when mapping differentially expressed genes to the human metabolic model Recon2. Clustering enrichment scores from the highest in red to the lowest in blue were calculated for different gene expression thresholds ($\text{Log}_2|\text{FC}|$) and percentages of retained currency metabolites. e Gene network corresponding to the maximal clustering enrichment score ($\text{Log}_2|\text{FC}|>3$; removed currency metabolites = 2%). The transcription of nodes in green was upregulated and those in red downregulated in Huh7-GCK⁺/HK2⁻ compared to Huh7 cells. Plain edges mark coregulation between nodes and broken edges an inverse regulation at the transcriptional level.

Minor comment 8

The last part of the result section is very adequate but is restricted to innate immunity. I propose that this be written in the heading and in the figure legends.

Reply

The paragraph, the heading and figure legend have been modified (Line 239).

Minor comment 9

In the discussion, comments on the mechanisms by which Hk2 expression increases in cirrhosis would be welcome.

Reply

HK2 increased expression during cirrhosis is now discussed in the discussion section - line 293-299 ,“ While GCK, but not HK2, is expressed in normal hepatocytes, the expression of HK2 occurs during cirrhosis and increases as the disease progresses to carcinoma. Several signaling pathways such as hypoxia inducible factors (HIF), peroxisome proliferator-activated receptors (PPAR) and phosphatidylinositol-4,5-bisphosphate 3-kinase (PI3K) might contribute to HK2 induction in fatty liver disease and its evolution towards cirrhosis and carcinogenesis (35-37). Consequently, HK2 induction has been proposed as a risk marker of HCC development (16)....”

Minor comment 10

Since citrate is an essential substrate of lipogenic pathway, have the authors explored the possibility that citrate be provided by the cell culture medium?

Reply

DMEM used for cell culture does not contain citrate. Therefore, the increase of intracellular lipids in Huh7-GCK⁺/HK2⁻ cells cannot be the result of an uptake of exogenous citrate.

Minor comment 11

The authors write that PC activity can be regulated by alternative mechanisms depending on HK isoenzyme expression: can this be further detailed?

Reply

The endometabolome analysis revealed that oxalate was decreased in Huh7-GCK⁺/HK2⁻ (Fig. 5c). This metabolite is a final degradation product of glyoxylate and is an inhibitor of liver PC, resulting in reduced neoglucogenesis and lipogenesis (O'Neill I. E. et al., 1984, Yoo J.-J. et al., 2019). Thus, a lower level of oxalate in Huh7-GCK⁺/HK2⁻ is likely to contribute to increased PC activity in these cells.

This now appears in the discussion section, lines 366-368, "Consistently, it was found that oxalate inhibits liver PC, resulting in reduced gluconeogenesis and lipogenesis (42-43). Thus, a higher PC activity could be explained by lower levels of oxalate in Huh7-GCK⁺/HK2⁻ cells."

Reviewer #2:

Major comments

1: The authors write in the abstract that a switch from GCK to HK2 is occurring during the transition from primary hepatocytes to hepatocellular carcinoma. It is therefore counterintuitive that the authors find that 250 HCC tumors display high expression of GCK while only 115 have low expression. Further, although the Spearman correlation test was performed to assess the correlation between HK2 and GCK expression, the correlation coefficient is not indicated. For the analysis in Fig. 1 to be any meaningful, it is critical to compare the same patients displaying either high and or low expression of the studied enzymes. The optimal stratification method used in Fig 1 does not add much and leads to incorrect conclusion.

*Defining the tumors according to their hexokinase low and high expression level was indeed inappropriate and misleading. Lowest and highest expression was the proper definition. This was corrected - lines 106-108, "Although HK1 or HK3 expression level were not associated to patient survival rate (Fig. 1a), **highest** expression levels of HK2 as previously described (19) and **lowest** expression levels of GCK in the tumors were associated with a lower survival rate."*

In the previous version of the manuscript, a dot-graph showed the inverse correlation between GCK and HK2 expression for all patients for whom transcriptome data were available (n=365). Spearman's analysis was performed but the correlation coefficient was indeed missing in the figure. The calculated r and p-value were -0.122 and 0.019, respectively, so that the inversely correlated expression of GCK and HK2 was statistically significant.

*To further characterize the discriminating aspect of these two isoenzymes on survival, we calculated the survival correlation coefficient related to their expression on the subset of 130 patients for whom the period between diagnosis and death is precisely known (uncensored data). The Spearman's coefficient is positive for comparison of overall survival and GCK expression level whereas it is negative for HK2. In addition, we confirmed that GCK and HK2 expression are inversely correlated in this group of 130 patients, and we provide associated r and p-values in the new Fig. 1C. This analysis is discussed in lines 114-119, "**Finally, correlation coefficients between patient survival in months and HK2 or GCK expression level were determined. For this, we only considered the subset of 130 patients for whom the period between diagnosis and death is precisely known (uncensored data), and performed a Spearman's rank correlation test (Fig. 1c). Patient survival was positively correlated to GCK expression but inversely correlated to HK2 expression in line with the Kaplan-Meier analysis. In addition, GCK and HK2 expression tends to be inversely correlated in tumor samples (Fig. 1c).**"*

For overall survival, HK2 expression is more discriminant than GCK expression but both represent statistically significant prognostic markers. To further support this statement, we introduced a supplementary panel in the figure 1 (Fig. 1b) showing that the GCK/HK2 ratio outperforms individual markers to stratify patients. Indeed, when patients were stratified on the basis of HK2 or GCK expression, the median survival between the corresponding subgroups differed by 33.8 and 36.5 months, respectively (Fig. 1a). This difference reached 42.8 months when the stratification of patients was based on the GCK/HK2 ratio (new Fig. 1b).

*In addition to new Fig. 1b and 1c, modifications appear as follow in the revised manuscript - lines 108-119, "**We thus stratified patients based on the GCK/HK2 expression ratio to combine these two markers (Fig. 1b). When patients were stratified on the basis of HK2 or GCK expression levels, the median survival between the corresponding subgroups differed by 33.8 and 36.5 months, respectively (Fig. 1a). This difference reached 42.8 months when the stratification of patients was based on the GCK/HK2 ratio (Fig. 1b). This demonstrated that the GCK/HK2 ratio outperforms HK2 or GCK expression alone as predictor of patient survival. Finally, correlation**"*

coefficients between patient survival in months and HK2 or GCK expression level were determined. For this, we only considered the subset of 130 patients for whom the period between diagnosis and death is precisely known (uncensored data), and performed a Spearman's rank correlation test (Fig. 1c). Patient survival was positively correlated to GCK expression but inversely correlated to HK2 expression in line with the Kaplan-Meier analysis. In addition, GCK and HK2 expression tends to be inversely correlated in tumor samples (Fig. 1c)."

Finally, patients were stratified in Fig. 1a based on individual expression levels of each HK, into two subgroups with the highest and lowest expression. Therefore, and depending on which HK is considered for stratification, one patient will fall either in the highest expression or the lowest expression subgroup. Besides, there is no rationale for having subgroups of equal size when performing this kind of analysis. For each HK, the subgroups yielding the lowest p-value when analyzing survival outcome are presented. This makes the stratification presented in Fig. 1a optimal for each HK as demonstrated by Uhlen M. & al. (Uhlen M. et al. 2017). This corresponds to the current standard used in Protein Atlas for TCGA data analysis, that has recently been used for HK2 stratification by Yoo et al. (Yoo J.-J. et al. 2019).

Figure 1. Correlation between hexokinase expression levels in HCC tumors and patient survival. **a** Kaplan–Meier estimates of the survival of HCC patients depending on the expression of HK1, HK2, HK3 and GCK (HK4) genes in tumor biopsies (n=365; TCGA expression data retrieved from cBioPortal) (70, 71). Duplicate analyses from the same patient were removed as well as patients who died when biopsied (overall survival=0 months). Optimal stratification based on highest and lowest gene expression values was determined using Protein Atlas database (18). **b** Same as above but patients were stratified based on the GCK/HK2 gene expression ratio. The stratification showing the lowest p value when comparing subgroups of patients with the highest to the lowest GCK/HK2 expression ratio is displayed. Patient TCGA-DD-AAE9 exhibiting undetectable levels of GCK and HK2 was removed from this analysis as the GCK/HK2 ratio could not be calculated. **c** Correlations between patient survival, GCK expression and HK2 expression. Spearman’s rank correlation test on the subset 130 patients for whom the period between diagnosis and death is precisely known (uncensored data).

2: The authors generate a HK2 knockout Huh7 cell line which is then replaced by GCK. How is this possible if HK2 is required to sustain proliferation of HCC cells (as stated by the authors in the abstract)? Do Huh7 cells proliferate faster than the GCK-HK2 line? If so, the data needs to be corrected and normalized for this. Further, only the Huh7 cell line is included in this study. Confidence would be significantly increased by including both primary hepatic cells as well as a cell line panel in the analysis. The HK1 and HK3 needs to be added to in Fig2A and it is unclear why HK activity was measured from frozen and not fresh material? Several controls are missing in Fig 2B, for instance knockdown of HK1 and HK3, as well as does re-introduction of GCK affect the expression of the other hexokinase 1-3?

The question of cell proliferation was also raised by the Reviewer 1. As aforementioned, we observed no differences in the proliferation of Huh7 or Huh7GCK⁺HK2⁻ cells. Cell counts were recorded at each passage of Huh7 and Huh7-GCK⁺/HK2⁻ cell cultures over the last 12 months and were used to calculate doubling time. Over 75 passages, the average doubling-time was not statistically different between the two cell lines (31.8±1.0 and 33.3±1.1 hours for Huh7 and Huh7-GCK⁺/HK2⁻ respectively). We also monitored their proliferation kinetic over 72h of culture and found no difference at any time (see Supplementary Fig. 1). This was mentioned in the revised version of the manuscript (line 132-133), “The cell proliferation capacity remained identical between the two cell lines (Supplementary Fig. 1).”

Supplementary Figure 1: Proliferation of Huh7 and Huh7-GCK⁺/HK2⁻. Cells were seeded in 24 well-plate under standard growth conditions and cellular proliferation was determined at time 0, 24, 48 and 72h post-seeding using the CellTiter-Glo®

Luminescent Cell Viability Assay (Promega). Luminescence were quantified with an Infinite M200 microplate reader (TECAN). Means \pm SEM are presented (n=3).

Huh7 were transduced for GCK expression prior to the invalidation of HK2 so that the cells express at least one hexokinase at all steps of selection. The proliferation data show that GCK and HK2 can both efficiently support the generation of intermediates of biosynthesis allowing cell proliferation.

In addition to Huh7 (hepatocarcinoma cell line), GCK was transduced in Huh6 (hepatoblastoma cell line) and Vero (kidney epithelial cell line). Bodipy staining showed that intracellular triglycerides accumulation was induced by the expression of GCK in both Huh7 and Huh6 cells but not in Vero cells, indicating that the expression of GCK increases lipogenesis in metabolically relevant cells, independently of the expression of HK2. Data now appear in Supplementary Fig. 5 and results are presented in lines 168-172, “The accumulation of neutral lipids in Huh7 expressing both HK2 and GCK indicates that the phenotype is driven by GCK expression and not by HK2 knockdown (Supplementary Fig. 5). Lipid accumulation upon GCK expression was also observed in Huh6 hepatoblastoma cells but not in epithelial kidney Vero cells, indicating that this phenomenon occurs in metabolically relevant cells (Supplementary Fig. 5).”

Supplementary Figure 5. Intracellular lipid content of two hepatic and one renal cell lines was analyzed after stable re-expression of GCK. Parental Huh6, Huh7 or Vero cells were transduced with lentiviruses for GCK expression using a pLEX-GCK construct as described in material and methods. Cells were then cultured for 7 days in the presence of puromycin to select transduced cells before amplification. Cells were stained for their intracellular lipid content using BODIPY 493/503 dye and analyzed by flow-cytometry. Means \pm SEM of fluorescence normalized to the corresponding parental cell line are presented.

GCK-induced TG in Huh6 but not in Vero cells was confirmed by TG quantification (Fig. R2).

Figure R2: Intracellular levels of triglycerides were quantified in Huh6 and Vero cells transduced for stable expression of GCK. TG were quantified after cell homogenization using a specific enzymatic assay (Fluorometric TG Quantification kit, Cell Biolabs). Presented data correspond to means \pm SEM.

Comparison of Huh7-GCK⁺/HK2⁻ cells with the parental Huh7 differing only for the expression of the hexokinase isoenzyme is a strong demonstrator that the sole re-expression of GCK in HCC can restore some of the essential functions of normal hepatocytes. Reviewer 2 also suggested that engineering primary human hepatocytes (PHH) would reinforce the observation. PHH already express GCK and the engineering here would be to knockdown GCK after transduction of HK2. Although PHH can be purchased at a prohibitive cost, they cannot be expended in cell culture and genetic manipulations are extremely challenging, and cannot be performed in a reasonable timeframe without important means. In addition, important variations from donor to donor would require repeating these experiments with a large set of donors.

Concerning HK1 and HK3 expression in Huh7 and Huh7-GCK⁺/HK2⁻ cells we are now showing their absence at the protein level by western blot (See Fig. 2a in the revised manuscript), in line with the absence of RNA in the transcriptome data, ruling out the rationale for HK1 and HK3 knockdown in Huh7 or Huh7-GCK⁺/HK2⁻ cells. See line 126, “The exclusive expression of HK2 and GCK in Huh7 and Huh7-GCK⁺/HK2⁻ cell lines, respectively, was validated, while HK1 and HK3 were not expressed (Fig. 2a).”

Finally, the enzymatic activity of HKs was performed on frozen material (-80°C) for practical reasons and normalization of our experimental procedures. The procedure for sample preparation has been clarified in the “material and methods” section of the revised manuscript - line 447-448, “...Cells were trypsinized, washed twice, and cell pellets were stored at -80°C. Protein extractions and assays were performed as previously described...”

3: The Huh7-GCK⁺/HK2⁻ cells showed a higher migratory capacity with lipid droplet accumulation. As cell migration plays an important role in tumor metastasis. Lipid accumulation is also increased in different neoplastic processes with undergoing cell proliferation. Again, it would be required to assess the cell proliferation. Secondly, this seems to be in conflict with the conflict to the Kaplan–Meier results. Needs clarification.

As indicated above, the proliferation capacity of Huh7-GCK⁺/HK2⁻ and Huh7 is similar. The motility test was performed to demonstrate that the other function of GCK, that is regulation of cell migration (Kishore M. et al., 2017), was also restored in Huh7GCK⁺/HK2⁻ cells. It also confirmed the Ingenuity analysis highlighting cellular movement as a modified function following the isoenzyme switch. Although suggested by the data, the hypothesis

that GCK-expressing cells have a higher migratory capacity in vivo needs extensive additional work to be demonstrated.

If the hypothesis revealed to be true, it does not necessarily conflict with the Kaplan-Meier analysis because Huh7-GCK+/HK2- cells have a restored innate immunity and sensitivity to NK cells cytotoxicity (Fig. 6). NK cells play an essential role in the immune surveillance of tumors and elsewhere, the activation of the innate immunity in cancer immunotherapy, has now been clearly identified as a promising strategy to render cancer cells susceptible to NK-mediated cytotoxicity (Ben-Shmuel A. et al., 2020). It can thus reasonably be assumed that HCC expressing GCK are more susceptible to NK cells, allowing their elimination and a better outcome even if a higher migratory capacity is suspected. On the contrary, HCCs expressing HK2 being resistant to NK cells, they would more easily escape tumor immune surveillance. This hypothesis needs to be validated and will be the subject of future studies.

4: For next generation sequencing data analysis, the Student's t-test with Benjamini-Hochberg correction was used to find the differentially expression genes. However, considering that the distribution of the data is not that expected by a Student's t-test. A more appropriate method should be used incl. EdgeR or DESeq2 on the raw counts to identify the correctly identify differentially expressed genes.

We reanalyzed NGS data using DESeq2 method and modified the figures accordingly (Fig 2c-e, 3a-c, 5j and 6a-b). This new analysis largely overlaps the previous one and reinforces the initial observations. The number of genes whose expression varies in the enrichment analysis, and which are involved in "cellular movement" and "lipid metabolism", slightly increases thus reinforcing the p-values.

5: One important function of hepatocytes is to secrete triglyceride-rich VLDL. In contrast, the HCC cells produce less VLDL. Again, the data of the primary hepatic cells should be included to have a proper control. Indeed, HCC cells loss their capacity to produce and secrete bona-fide VLDL.

It is well known that Huh7 cells do not produce VLDL but ApoB+ lipoproteins of LDL density (Meex S. J. R. et al., 2011). In contrast, density of ApoB+ lipoproteins secreted by Huh7-GCK⁺/HK2⁻ match the density of VLDL secreted by PHH (Beilstein et al. 2017) and of circulating VLDL in vivo (Ginsberg H. N., 1998). This is now clarified in the text - lines 186-188, "As expected, lipoproteins secreted by Huh7 sediment at the density of LDL, while those secreted by Huh7-GCK⁺/HK2⁻ (Fig. 4d) match the density of VLDL found in human plasma or secreted by primary human hepatocytes in culture (26, 27)."

6: To study a snap shot of the metabolites of one cell line with or without genetic modifications does not add much. A proper isotopomer flux analysis needs to be performed to resolve the metabolic reprogramming. It is also surprising that the authors find that the Huh7-GCK+/HK2- cells have modified TCA-cycle and produce more intracellular metabolites compared to Huh7 cells. Cancer cells (which is here more represented in Huh7 cell line) are expected to produce more metabolites to support their growth. Again, this can easily be resolved by performing isotopomer analysis. Although Huh7-GCK+/HK2- cells have higher sensitivity to RIG-I and MDA5 ligands, the authors completely lack to provide any insights into the connection from the rewired metabolism rewiring, ISRE induction, and the role of GCK in prolonged survival. This needs to be resolved.

We undertook a fluxomic analysis of glycolysis by culturing cells with [U-¹³C] glucose and analyzing ¹³C-pyruvate as an end product of glycolysis. M+3 labeled pyruvate produced from [U-¹³C] glucose was significantly more important in Huh7-GCK⁺/HK2⁻ cells compared to Huh7 cells and is now presented in Fig. 5f. In addition, new data showing that glucose consumption is increased while lactate secretion is decreased in Huh7-GCK⁺/HK2⁻ cells are now presented (Fig. 5d and e). Altogether these news observations point to an increased glycolytic flux in Huh7-

GCK⁺/HK2⁻ cells and a differential usage of pyruvate preferentially entering the TCA rather than generating lactate. The manuscript has been modified according to these new observations - lines 205-210, "...This led to investigate glucose catabolism in further details. Glucose consumption and stable isotope incorporation from [U-¹³C]-glucose into pyruvate were both increased in Huh7-GCK⁺/HK2⁻ cells compared to Huh7 cells (Fig. 5d and f). This increased glycolytic flux together with a reduced lactate secretion (Fig. 5e) is likely to account for the elevation of lactate levels and suggest that the increased pyruvate production essentially fuels mitochondrial TCA cycle in Huh7-GCK⁺/HK2⁻ cells." A complete analysis using different ¹³C-substrates metabolites will certainly help in resolving the observed metabolic reprogramming, but we believe this is the matter of another article requiring important human, technical and financial resources.

To address the second point raised by Reviewer #2, we have inserted a new paragraph in the discussion section where interactions between RIG/MDA5 signaling and metabolism rewiring are discussed – lines 388-394 "This immune phenotype occurs in a context of reduced SDH activity and increased intracellular content in succinate (Fig. 5l-m). A pro-inflammatory function of immune cells such as macrophages was previously linked to TCA rewiring, with reduced SDH activity resulting in succinate accumulation (39, 50, 51). Succinate can also be secreted from LPS-activated macrophages and activate its cognate receptor, succinate receptor 1 (SUCNR1, previously known as GPR91) in an autocrine and paracrine manner to further enhance production of IL-18 (52)." and - lines 399-402, "...Here we show that HK2 knockdown promotes RIG-I-induced ISRE-dependent transcription (Fig. 6d). This is consistent with the results obtained by Zhang W. et al. (48), indicating that HK2 interaction with MAVS restrains RIG-I-induced IFN- β secretion."

7: The transcriptomic data with the GO term "Type I IFN signaling pathway" was analyzed, however, instead of mentioning just one GO term, the authors should show the results from GO enrichment analysis with significance evaluation and explain why focus on this GO term.

A GO enrichment analysis has been performed on up-regulated and down-regulated transcripts in Huh7-GCK⁺/HK2⁻ vs Huh7 cells and is shown in the new supplementary Figure 8 for the Top 10 enriched GO-terms. Type I interferon signaling pathway appears as a significantly enriched GO term in up-regulated transcripts, together with other innate immunity pathways (regulation of complement activation, regulation of inflammatory response, interferon-gamma-mediated signaling pathway, cellular response to lipopolysaccharide, response to cytokine and response to lipopolysaccharide). In contrast, no innate immunity-related pathways were enriched in the down-regulated transcripts analysis. Therefore, we have focused the analysis on Type I interferon signaling pathway, for which a detailed analysis is shown in Fig. 6a-b (updated after DESeq2 analysis). The paragraph "Restored innate immune sensitivity in Huh7-GCK⁺/HK2⁻" in the "results" section has been modified accordingly lines 243-254 "The functional analysis of gene ontology (GO) terms associated to differentially expressed transcripts revealed an enrichment in terms related to the regulation of innate immunity. The gene signature associated with type-I interferon (IFN) signaling pathway scored among the top enriched terms of upregulated genes in Huh7-GCK⁺/HK2⁻ cells (Supplementary Fig. 7). Within the 91 gene members of this GO term, 20 transcripts of Type I-IFN signaling were significantly up-regulated in Huh7-GCK⁺/HK2⁻ compared to Huh7 (Fig. 6a). This includes interferon regulatory factors (IRF1, IRF7 and IRF9), IFN-stimulated genes (ISGs) such as ISG15, MX1, OAS1, OAS3, RNaseL and signaling intermediates such as IKBKE coding for IKK ϵ (Fig. 6b). The chaperon HSP90AB1, which is involved in the phosphorylation and activation of STAT1, was also induced (33). In contrast, two genes were significantly down-regulated in Huh7-GCK⁺/HK2⁻ compared to Huh7, the RNaseL inhibitor ABCE1, and TRIM6, an E3 ubiquitin-protein ligase regulating IKK ϵ ."

Supplementary Figure 7: GO enrichment analysis of transcripts significantly up- and down-regulated in Huh7-GCK⁺/HK2⁻ vs Huh7 cells determined by DESeq2 analysis of transcriptomic data. Transcripts lists were submitted to the Functional Annotation Tool of the online knowledge base DAVID Bioinformatics Resources 6.8, NIAID/NIH. In DAVID, Fisher's Exact test is adopted to measure the gene-enrichment in annotation terms. Parameters were adjusted to a minimal count belonging to the annotation of 10 and EASE score (one-tail Fisher Exact Probability Value) threshold of 0.01.

8: It is surprising that succinate accumulates in the Huh7-GCK⁺/HK2⁻ and simultaneously have high respiration. Is Succinate dehydrogenase mutated in the cell line? Authors propose that other pathways maintain a high activity of the respiratory chain by fueling complex III but this was never tested. It is necessary to repeat the seahorse data and only provide cells with substrates that feeds into complex 1-3, respectively and re-do the assay in order to make such conclusion.

Activity of complex II was measured *ex-cellulo* after immunopurification. In these conditions, the intrinsic SDH activity was identical between the two cell lines, indicating that SDH is not mutated and is fully functional in Huh7-GCK⁺/HK2⁻ in the absence of intracellular regulation (Fig. R3 and Fig. 5m).

Figure R3: The functionality of SDH in Huh7 and Huh7-GCK⁺/HK2⁻ cells was analyzed with the Complex II Enzyme Activity Microplate Assay Kit (ref ab109908, Abcam). Complex II was extracted from Huh7 and Huh7-GCK⁺/HK2⁻ cells accordingly with the manufacturer instructions. After normalization of cell extracts for protein concentrations, Complex II was immunocaptured onto a microwell plate and the SDH activity of the immobilized complex was determined.

Additional seahorse experiments were performed confirming that Huh7-GCK⁺/HK2⁻ cells display a higher mitochondrial respiration compared with Huh7, with an increased basal and maximal respiration, ATP production and spare respiration capacity (now in Fig 5n, o and Supplementary Fig. 6 also presented below). By sequential inhibition of Complex I and III of the respiratory chain, we found that OCR is mainly dependent on Complex I activity which is fueling Complex III (Fig 5n and 5o). In Huh7-GCK⁺/HK2⁻ with a reduced SDH activity (Fig. 5m), increased respiration remains mainly dependent on Complex I (80%) although contribution of complex III slightly increased to 7% (vs 4% in Huh7). These results have been introduced in Fig. 5, and in the main text - lines 232 to 238, “...we observed that the overall oxygen consumption was increased in Huh7-GCK⁺/HK2⁻ (Fig. 5n) with increased basal and maximal respiration, ATP production and spare respiration capacity (Supplementary Fig. 6). Functional analysis of the respiratory chain showed that oxygen consumption in Huh7 and Huh7-GCK⁺/HK2⁻ cells was mainly dependent on complex I which is fueling complex III (Figs. 5n, o). Thereby, the HK isoenzyme switch rewired the TCA cycle promoting carboxylation of pyruvate into OAA in the presence of a reduced SDH activity and increased respiration through complex I.”

A comparative analysis of basal respiration, maximal respiration, ATP production and spare respiration capacity calculated from OCR data analysis is now presented in supplementary Figure 6.

Figure 5: n Oxygen consumption rate (OCR) in Huh7 and Huh7-GCK⁺/HK2⁻ cells were determined with a Seahorse analyzer before and after the addition of oligomycin (Complex V inhibitor), FCCP (uncoupling agent), rotenone (Complex I inhibitor) and antimycin A (Complex III inhibitor).

Figure 5: o Non-mitochondrial, complex I-dependent and complex III-dependent maximal OCR were calculated from (N).

Supplementary Figure 6 : Basal respiration, Maximal Respiration, ATP production and Spare Respiration Capacity calculated from OCR data generated with Seahorse analyzer (cf. Fig. 5N), as preconized in Seahorse XF cell Mito Stress Test. The results are presented as means \pm SEM (n=5).

Minor:

8: Please define the R value in Figure 1B

This panel has been removed in the novel version of the manuscript.

9: Please define the P value in Figure 4G

Figure 4g is now figure 5j. The p-values are now presented on the graph.

10: Specify adjust p value instead of p value

Done.

11: Revise the Scientific notation in the method part (e.g. 1.105 cells per well, 3.105 or 3.106 NK cells etc.)

Done.

12. Many critical experimental details are missing, for instance which plates (company, cat no) were used for the migration assay? How were the picture taken? Microscope? If so, objective? Etc. etc.

Done.

REVIEWERS' COMMENTS:

Reviewer #1 (Remarks to the Author):

The authors have adequately responded to the previous comments made.
The information provided in this revised version is adequate, significant and meaningful.
I consider that the manuscript can be published without further review

Reviewer #2 (Remarks to the Author):

The authors have added a significant amount of new data and have addressed most of my previous comments. The overall quality of the manuscript has greatly improved.

I only have one minor comment on the conclusion on Fig 5N which I disagree with. I feel this is important to address.

'Functional analysis of the respiratory chain showed that oxygen consumption in Huh7 and Huh7-GCK+/HK2-238 cells was mainly dependent on complex I which is fueling complex III'

As Rotenone (complex I inhibitor) was injected prior Antimycin A (complex III inhibitor), it is likely that Rotenone already completely blocks respiration, so the subsequent injection of Antimycin A have consequently no effect/or little effect.

I would therefore strongly suggest to repeat the very same experiment but now inject Antimycin A prior Rotenone -this would allow the authors to make a proper conclusion of that experiment.